# OSCAR: Orthogonalized Sequential Component Analysis for Tensor-on-Tensor Regression

## Abstract

Tensor-on-tensor (TOT) regression is a critical task in many fields. However, its application is severely hindered by the "curse of dimensionality" arising from the exponential growth of parameters in the coefficient tensor. Existing methods primarily fall into two categories: low-rank approximations, which often have limited predictive accuracy and interpretability, and sequential component extraction methods that rely on data-space deflation. This deflation mechanism suffers from greedy sub-optimal solutions, error propagation, and a lack of component orthogonality, hindering feature disentanglement. To address these limitations, we propose **O**rthogonalized **S**equential **C**omponent **A**nalysis for Tensor-on-Tensor **R**egression (**OSCAR**). First, we design an Input-Mode Orthogonal Block Term (**IMOBT**) low-rank structure for the coefficient tensor, which inherently enables the supervised extraction of orthogonal components. Building on this, we develop a Sequential Riemannian Optimization (**SRO**) framework that replaces classical data-space deflation with explicit geometric constraints in the parameter space. This is achieved through a Subspace Constraint Riemannian Gradient Descent algorithm on a Stiefel manifold to rigorously enforce orthogonality. Furthermore, to alleviate the greedy bias of sequential learning, we introduce a novel collaborative refinement mechanism that re-optimizes the synergy among all components whenever a new one is added, enabling an iterative "look-back" for a superior global solution. Extensive experiments on synthetic and real-world datasets demonstrate that our proposed OSCAR framework not only achieves competitive predictive performance but also shows significant advantages in supervised component extraction and feature disentanglement.

## 1 Introduction

The structured analysis of tensor (multi-way array) data is one of the core challenges in modern statistical regression (Kolda & Bader, 2009). Data are often presented in the form of tensors in many fields, such as neuroscience (Zhou et al., 2013), recommendation systems (Zhu et al., 2018), and clinical applications (Wang et al., 2014). Modeling the multi-modal structure of tensors is a critical problem. Early work mostly addressed scalar-on-tensor (Chen et al., 2019; Hou et al., 2015) or vector-on-tensor regression problems (Miranda et al., 2018). In recent years, methods for tensor-on-tensor (TOT) regression have begun to be explored. Tensor-on-tensor regression directly models this type of data by preserving the intrinsic structures of both the covariates and the response variables. However, the coefficient tensor in an unconstrained TOT model grows exponentially with its order, leading to the severe "curse of dimensionality" (Donoho et al., 2000). This poses significant challenges in computation, storage, and a high risk of overfitting.

To address this challenge, methods based on low-rank approximation of the regression coefficient tensor have been proposed. This paradigm assumes that the entire coefficient tensor $\mathcal{B}$ possesses an intrinsic low-rank structure and introduces various tensor decompositions to enforce this low-rank assumption. Initial studies focused on the CANDECOMP/PARAFAC (CP) low-rank assumption, which represents the coefficient tensor as a sum of rank-one tensors (Lock, 2018). The Tucker low-rank assumption decomposes the tensor into a core tensor and a set of factor matrices along each mode (Li et al., 2018; Gahrooei et al., 2021). Tensor Train (TT) decomposition represents a high-order tensor as a chain of matrix products of third-order core tensors (Liu et al., 2020; Qin & Zhu, 2025). Although these methods approximate the regression tensor from different perspectives, they often provide little insight into the relationships between the covariates $\mathcal{X}$ and the response $\mathcal{Y}$.

Another line of research focuses more on extracting components that are important for the regression based on Partial Least Squares (PLS) (Wold, 1975). The core idea of this paradigm is to greedily and sequentially extract components that maximize the covariance between the input $\mathcal{X}$ and the output $\mathcal{Y}$. N-way partial least squares (N-PLS) decomposes the predictor $\mathcal{X}$ and response $\mathcal{Y}$ into a series of rank-one tensors, aiming to maximize the pairwise covariance between their score vectors (Bro, 1996). To ensure the separation of each latent variable, it introduces data-space deflation after

each iteration, which involves subtracting the structure explained by the extracted component from the original data. The next component is then extracted from the resulting residual. However, the rank-one assumption of N-PLS, based on CP decomposition, limits its modeling capacity. To this end, High order partial least squares (HOPLS) upgrades the definition of latent variables from strictly rank-one vectors to a more flexible Tucker subspace, allowing for the extraction of richer structured information in certain modes (Zhao et al., 2012).

Despite this improvement, both N-PLS and HOPLS rely on data-space deflation. This classical deflation method has inherent limitations: (1) The greedy nature, while greatly simplifying the optimization process, can lead the model to a sub-optimal solution because early components cannot be revised by later discoveries. This also leads to error propagation and accumulation, which affects the extraction and learning of subsequent latent variables. (2) These methods rely on the computation of covariance between tensors, a step with high complexity that hinders scalability to large datasets. Moreover, they do not directly optimize for the reconstruction error, often resulting in limited predictive performance. (3) The factor matrices for different components within the same mode are not orthogonal, leading to insufficient feature disentanglement and weaker interpretability.

In summary, existing tensor regression methods either solely consider low-rank approximations of the regression tensor or focus exclusively on extracting components supervised by a covariance objective. How to combine low-rank approximation with supervised component extraction remains unsolved. In this work, we propose a novel low-rank approximation structure that not only avoids the curse of dimensionality in tensor regression but also enables the supervised extraction of orthogonal components, resulting in a more accurate and interpretable regression framework.

First, we design an ingenious low-rank structure of the regression coefficient tensor. Inspired by the Tucker product structure as a form of higher-order principal component analysis (HOPCA), we treat the factor matrices as feature extraction directions and the resulting core tensor as the corresponding components. Based on this, we devise a novel low-rank structure for the regression tensor, as shown in Figure 1. Our low-rank structure not only reduces the number of parameters in tensor regression, but also directly uses the reconstruction error as its supervision signal. The factor matrices of different components in the same mode are set to be orthogonal, thereby extracting a set of orthogonal components.

Furthermore, we adopt a sequential extraction framework for optimization, allowing us to progressively extract the most important components for prediction. Instead of using data-space deflation, we impose explicit geometric constraints in the parameter space to achieve a more rigorous and collaborative process for component extraction and optimization. Specifically, we impose strict column-space orthogonality constraints on the factor matrices within the same mode, without modifying the data. This constraint directly ensures at the parameter level that the "feature extraction subspaces" of different components are completely separate. This stronger constraint achieves a more principled form of feature disentanglement and also serves as an effective regularization, reducing model complexity. We introduce a Riemannian Gradient Descent (RGD) optimization method tailored for our task (Absil et al., 2008; Qin & Zhu, 2025; Luo & Zhang, 2024). The optimization of a factor matrix $\mathbf{B}$ is performed on a subspace constrained Stiefel manifold, which consists of all matrices satisfying $\mathbf{B}^T\mathbf{B} = \mathbf{I}_k$ and being orthogonal to the subspace spanned by previous feature extracting matrices of the same mode (i.e., $\mathbf{B}^T\mathbf{Q} = 0$). We employ a subspace constrained Riemannian Gradient Descent (RGD) algorithm to rigorously and efficiently solve this constrained optimization problem.

Finally, to overcome the purely greedy nature of traditional sequential methods, we introduce a Cooperative Refinement Mechanism. When computing a new component in a stage-wise manner, the model not only optimizes this new component but also re-optimizes a synergy matrix $\mathbf{A}$ for all previously existing components. This allows the established component structures to perceive the addition of a new member and dynamically adjust their contributions, enabling all components to work together to achieve optimal overall performance. This "look-back refinement" capability endows our sequential learning process with an iterative global consideration, aiming to compensate for the short-sightedness of traditional deflation methods.

We believe this framework provides a novel perspective for sequential tensor regression. The main contributions are:

**A New Paradigm for Sequential Tensor Regression.** We propose a novel low-rank decomposition based on geometric constraints in the parameter space, which can generate a set of orthogonal principal components.

**A Stage-wise RGD-based Optimization Scheme.** We design a rigorous geometric optimization framework that enables us to sequentially identify components that are important for prediction along orthogonal projection directions. A Subspace Constrained Riemannian Gradient Descent (SCRGD) algorithm is proposed to ensure the model strictly maintains its required geometric structure.

Figure 1: Framework of OSCAR-IMOBT

**A Novel Collaborative Refinement Mechanism.** We present a collaborative refinement mechanism to mitigate the greedy nature of sequential learning. By re-optimizing the synergy among all components whenever a new one is introduced, the model performs iterative "look-back refinement", leading to a superior global solution.

We have conducted extensive experiments on both synthetic and real-world datasets. The results demonstrate that our OSCAR framework not only surpasses existing state-of-the-art (SOTA) methods in predictive performance but also shows unique advantages in model interpretability and feature disentanglement.

## 2 PRELIMINARIES

In this section, we establish the notations and introduce the fundamental concepts required for understanding our proposed method. We begin with tensor fundamentals, define the tensor-on-tensor regression problem, review classical tensor decompositions, and finally, introduce the geometric framework for optimization under orthogonality constraints.

### 2.1 NOTATIONS AND TENSOR FUNDAMENTALS

We denote scalars by lowercase letters ($a$), vectors by bold lowercase letters ($\boldsymbol{v}$), matrices by bold uppercase letters ($\mathbf{M}$), and tensors by calligraphic letters ($\mathcal{X}$). For a tensor $\mathcal{X} \in \mathbb{R}^{I_1 \times \cdots \times I_N}$ of order $N$, its $(i_1, \ldots, i_N)$-th element is $x_{i_1, \ldots, i_N}$.

- **Tensor Contracted Product:** Let $\mathcal{X} \in \mathbb{R}^{d_1 \times \cdots \times d_n}$ be a tensor of order $n$ and $\mathcal{B} \in \mathbb{R}^{d_1 \times \cdots \times d_n \times o_1 \times \cdots \times o_m}$ be a tensor of order $n + m$. Their contraction over the first $n$ shared modes results in a tensor $\mathcal{Y} \in \mathbb{R}^{o_1 \times \cdots \times o_m}$ of order $m$. The elements of $\mathcal{Y}$ are computed as:

$$y_{i_1, \ldots, i_m} = \langle \mathcal{B}, \mathcal{X} \rangle_{1, \ldots, n} = \sum_{j_1=1}^{d_1} \cdots \sum_{j_n=1}^{d_n} b_{j_1, \ldots, j_n, i_1, \ldots, i_m} x_{j_1, \ldots, j_n}. \tag{1}$$

The Frobenius norm of a tensor $\mathcal{X}$ is defined as $||\mathcal{X}||_F = \sqrt{\sum_{i_1, \ldots, i_n} x_{i_1, \ldots, i_n}^2}$. Due to the curse of dimensionality of $\mathcal{B}$, most tensor regression approaches assume $\mathcal{B}$ has a low-rank structure. The mathematical definitions of the CP, Tucker, and Tensor Train low-rank structures can be found in Appendix F.

- **i-mode Product:** The $i$-mode product of a tensor $\mathcal{X} \in \mathbb{R}^{d_1 \times \cdots \times d_n}$ with a matrix $\mathbf{M} \in \mathbb{R}^{d_k \times r}$ is a tensor of size $d_1 \times \cdots \times r \times \cdots \times d_n$, denoted as $\mathcal{X} \times_i \mathbf{M}$.

## 2.2 TENSOR-ON-TENSOR REGRESSION

The Tensor-on-Tensor (TOT) regression task seeks to find a linear relationship between a predictor tensor $\mathcal{X} \in \mathbb{R}^{d_1 \times \cdots \times d_n}$ and a response tensor $\mathcal{Y} \in \mathbb{R}^{o_1 \times \cdots \times o_m}$. The model is formulated as:

$$\mathcal{Y} = \langle \mathcal{B}, \mathcal{X} \rangle_{\{1,\cdots,n\}} + \mathcal{E}, \tag{2}$$

where $\mathcal{B} \in \mathbb{R}^{d_1 \times \cdots \times d_n \times o_1 \times \cdots \times o_m}$ is the coefficient tensor and $\mathcal{E}$ is the noise tensor. The primary challenge is the "curse of dimensionality" associated with estimating $\mathcal{B}$. Given a dataset of $N$ samples $\{(\mathcal{X}_s, \mathcal{Y}_s)\}_{s=1}^N$, the objective is to find a $\mathcal{B}$ that minimizes the sum of squared errors:

$$\min_{\mathcal{B}} \sum_{s=1}^N ||\mathcal{Y}_s - \langle \mathcal{B}, \mathcal{X}_s \rangle_{\{1,\cdots,n\}}||_F^2. \tag{3}$$

## 2.3 OPTIMIZATION ON MANIFOLDS FOR ORTHOGONALITY CONSTRAINTS

Our method directly enforces orthogonality constraints by performing optimization on the geometric space of matrices with orthonormal columns. (Edelman et al., 1998)

- **The Stiefel Manifold:** The set of all $n \times b$ matrices ($b \leq n$) with orthonormal columns forms the Stiefel manifold:
$$\mathrm{St}(n, b) = \{\mathbf{M} \in \mathbb{R}^{n \times b} \mid \mathbf{M}^\top \mathbf{M} = \mathbf{I}_b\}. \tag{4}$$

Optimization on these manifolds requires computing a Riemannian gradient, which lies in the tangent space of the manifold at a given point, and then performing a retraction to map the updated point back onto the manifold, thus rigorously preserving the orthogonality constraint at every step.

## 3 THE OSCAR FRAMEWORK: ARCHITECTURE AND SEQUENTIAL RIEMANNIAN OPTIMIZATION

This section delineates the dynamic learning process of the Orthogonalized Sequential Component Analysis for Tensor-on-Tensor Regression (OSCAR) framework. We first formulate the regression problem and introduce our proposed INPUT-MODE ORTHOGONAL BLOCK TERM (IMOBT) structure for the coefficient tensor. We then detail the Sequential Component Extraction via Riemannian Optimization (SRO) procedure that learns the model's parameters by progressively extracting orthogonal components, and finally, we introduce a collaborative refinement mechanism to achieve a superior global solution. Our framework circumvents the "curse of dimensionality" by designing a novel low-rank representation for $\mathcal{B}$. OSCAR constructs $\mathcal{B}$ from a set of learnable, structured components. This approach can accomplish two tasks: **(1) Input-Mode Orthogonal Block Term (IMOBT) decomposition for low-rank regression** and **(2) Supervised sequential Component Interaction**.

## 3.1 INPUT-MODE ORTHOGONAL BLOCK TERM (IMOBT) DECOMPOSITION

We first present the Block Term Decomposition (BTD) for the regression coefficient tensor $\mathcal{B}$, formulated as follows:

$$\mathcal{B} = \sum_{k=1}^K \mathcal{W}_k \times_1 \mathbf{B}_{1,k} \times_2 \mathbf{B}_{2,k} \cdots \times_n \mathbf{B}_{n,k}, \tag{5}$$

where $\mathcal{W}_k \in \mathbb{R}^{b_1 \times \cdots \times b_n \times o_1 \times \cdots \times o_m}$, $\mathbf{B}_{i,k} \in \mathbb{R}^{d_i \times b_i}$, and $\times_i$ denotes the $i$-mode product. Under BTD, the regression framework can be expressed as:

$$\hat{\mathcal{Y}} = \langle \mathcal{X}, \mathcal{B} \rangle_{\{1,\cdots,n\}} = \sum_{k=1}^K \langle \mathcal{X}_{\text{core}}^{(\mathbf{k})}, \mathcal{W}_k \rangle_{\{1,\ldots,n\}}, \tag{6}$$

where $\mathcal{X}_{\text{core}}^{(\mathbf{k})} = \mathcal{X} \times_1 \mathbf{B}_{1,p_1}^\top \times_2 \mathbf{B}_{2,p_2}^\top \cdots \times_n \mathbf{B}_{n,p_n}^\top$.

We propose a novel perspective on BTD by interpreting the factor matrix $\mathbf{B}_{i,k}$ as a projection matrix. Specifically, it maps the information along the $i$-th mode of the input tensor $\mathcal{X}$ from the original $d_i$-dimensional space to a latent $b_i$-dimensional subspace. Consequently, through $K$ groups of such projections, we obtain $K$ components. $\mathcal{W}_k$ acts as a

fully connected layer linking the $k$-th component to the response tensor $\mathcal{Y}$. This structural interpretation demonstrates that the BTD architecture inherently possesses the capability for component extraction.

To further enhance *feature disentanglement*, we aim to enforce orthogonality among the projection directions of different components within the same mode, thereby preventing redundant feature extraction. On the other hand, a rigid orthogonality assumption may be restrictive. For instance, if the information along mode $i$ of the input tensor is concentrated entirely within a specific low-dimensional subspace, a single component might be insufficient to capture it comprehensively. In such scenarios, it is desirable for the projection direction to be shared across different components to fully exploit the information within that subspace.

To address this, we introduce a **combination tuple** $\mathcal{P}$ to unify these modeling paradigms. The cardinality of $\mathcal{P}$ corresponds to the number of principal components, $K$. Each component $k$ is uniquely associated with a combination index vector $\mathbf{p}(k) = (p_1, \ldots, p_n) \in \mathcal{P}$, indicating that the feature extraction matrix $\mathbf{B}_{i,p_i}$ is utilized for the $i$-th mode. For the sake of brevity, we write $\mathbf{p}(k)$ as $\mathbf{p}$ in the following. The OSCAR framework supports multiple interaction schemes by defining different sets of combination tuples $\mathcal{P}$, which impose block-wise sparsity on the core tensor of Tucker decomposition:

1. **Diagonal Interaction:** A parsimonious scheme that selects only the blocks on the main diagonal of the core tensor. It is defined by the index set $\mathcal{P}_{\text{diag}} = \{(p, p, \ldots, p) \mid p = 1, \ldots, P\}$, yielding $P$ active blocks.

2. **Full-Factorial Interaction:** A scheme that selects all blocks in the core tensor, capturing all possible cross-component interactions. It is defined by the full Cartesian product of indices, $\mathcal{P}_{\text{full}} = \{1, \ldots, P\}^n$, yielding $P^n$ active blocks. This is equivalent to a standard Tucker decomposition. (Tucker, 1966)

3. **Banded Interaction:** An intermediate scheme that interpolates between the two extremes. It selects a band of blocks, activating only those within a specified distance from the main diagonal. This distance becomes a tunable hyperparameter to control the trade-off between model parsimony and expressiveness.

A more formal mathematical definition is provided in the Appendix G. This motivates us to propose the **Input-Mode Orthogonal Block Term (IMOBT)** structure, as shown in Figure 1. For the $k$th component with combination $\mathbf{p} = (p_1, \cdots, p_n) \in \mathcal{P}$ and each input mode $i \in \{1, \ldots, n\}$, the **Feature Extraction Matrix** $\mathbf{B}_{i,p_i} \in \mathbb{R}^{d_i \times b_i}$ extracts information from the $i$-th mode of $\mathcal{X}$ and projects it onto a $b_i$-dimensional subspace. Its columns form an orthonormal basis ($\mathbf{B}_{i,p_i}^\top \mathbf{B}_{i,p_i} = \mathbf{I}$), tasked with identifying a subspace containing the $p_i$-th set of salient features within the $i$-th mode. The orthogonality between bases of different components ($\mathbf{B}_{i,p_i}^\top \mathbf{B}_{i,q} = \mathbf{0}$ for $q < p_i$) is the cornerstone of our framework.

For the $k$th component with combination $\mathbf{p} = (p_1, \ldots, p_n) \in \mathcal{P}$, a multilinear projection generates a corresponding core tensor:

$$\mathcal{X}_{\text{core}}^{(\mathbf{k})} = \mathcal{X} \times_1 \mathbf{B}_{1,p_1}^\top \times_2 \mathbf{B}_{2,p_2}^\top \cdots \times_n \mathbf{B}_{n,p_n}^\top \tag{7}$$

where $\mathcal{X}_{\text{core}}^{(\mathbf{k})} \in \mathbb{R}^{b_1 \times \cdots \times b_n}$. The resulting core tensors, $\{\mathcal{X}_{\text{core}}^{(\mathbf{k})}\}$, are concatenated along a new mode to form the final **Core Tensor** $\mathcal{X}_c \in \mathbb{R}^{b_1 \times \cdots \times b_n \times K}$. The final prediction $\hat{\mathcal{Y}}$ is obtained by contracting the feature tensor $\mathcal{X}_c$ with a learnable **Regression Weight Tensor** $\mathcal{W} \in \mathbb{R}^{b_1 \times \cdots \times b_n \times K \times o_1 \times \cdots \times o_m}$: $\hat{\mathcal{Y}} = \langle \mathcal{X}_c, \mathcal{W} \rangle_{\{1,\ldots,n+1\}}$.

Our feature extraction framework is compatible with different prediction heads, including the standard $L_2$-regularized fully connected (FC) layer as well as alternative low-rank tensor regression approaches such as CP, Tucker, or Tensor Train structure. Since the primary contribution of this work lies in the component extraction framework, we adopt the $L_2$-regularized FC layer as the prediction head in our experiments and theoretical analysis.

The OSCAR architecture implicitly defines the IMOBT structure for the global coefficient tensor $\mathcal{B}$. By tracing the flow of an input sample through the model, we can derive the explicit form of this tensor. The prediction $\hat{\mathcal{Y}}$ can be written as:

$$\hat{\mathcal{Y}} = \sum_{k=1}^{K} \langle \mathcal{X}_{\text{core}}^{(\mathbf{k})}, \mathcal{W}_k \rangle_{\{1,\ldots,n\}} \tag{8}$$

where $\mathcal{W}^{(\mathbf{k})}$ is the slice of $\mathcal{W}$ corresponding to the $k$th component. Substituting the definition of $\mathcal{X}_{\text{core}}^{(\mathbf{k})}$ and rearranging the contraction operations, we can express this in the canonical form $\hat{\mathcal{Y}} = \langle \mathcal{X}, \mathcal{B} \rangle_{\{1,\ldots,n\}}$, where the implicitly constructed $\mathcal{B}$ is:

$$\mathcal{B} = \sum_{\mathbf{k}=1}^{K} \mathcal{W}_k \times_1 \mathbf{B}_{1,p_1} \times_2 \mathbf{B}_{2,p_2} \cdots \times_n \mathbf{B}_{n,p_n}, \tag{9}$$

Figure 2: Illustration of the model framework. Left: For each component $k$, the model learns a set of mode-wise feature extraction matrices $\mathbf{M}_k = \{\mathbf{B}_{1,p_1}, \ldots, \mathbf{B}_{n,p_n}\}$, which are further refined into sequentially orthogonalized matrices $\mathbf{M}_k^{\text{sequential}} = \{\mathbf{B}_{1,p_1}, \mathbf{A}_{1,k}, \ldots, \mathbf{B}_{n,p_n}, \mathbf{A}_{n,k}\}$ to ensure orthogonal subspaces across components. Middle: Components are extracted one-by-one. At Step 1, the first component is fitted from the input tensor $\mathcal{X}$ to predict $\mathcal{Y}$. At Step 2 to Step $K$, previously extracted components are frozen (blue snowflake). Right: Visualization of the multi-$k$ structure, where higher-$k$ components encode progressively weaker but complementary information, forming an importance hierarchy across components.

where $\mathbf{B}_{i,p_i}$ denotes the feature extracting matrix for the $k$th component with combination $\mathbf{p} = (p_1, \cdots, p_n)$. Equation 9 reveals that OSCAR models $\mathcal{B}$ as a sum of structured, low-rank block terms, each built upon the effective projection operators. The orthogonality enforced on the spectral basis matrices $\{\mathbf{B}_{i,p_i}\}$ ensures that these blocks capture disentangled, non-redundant predictive information, giving the IMOBT structure its name and power. The optimization of the spectral basis matrices $\mathbf{B}_{i,p_i}$ is challenging due to the simultaneous orthonormality (intra-component) and mutual orthogonality (inter-component) constraints. To address this, we cast the optimization problem directly onto a geometric manifold where these constraints are inherently satisfied. The proof of parameter convergence for the IMOBT structure is detailed in Appendix B.

## 3.2 SEQUENTIAL ORTHOGONAL COMPONENT EXTRACTION AND COOPERATIVE REFINEMENT MECHANISM

Having defined the IMOBT architecture, we now elucidate its dynamic learning process. We have devised a **Sequential Riemannian Optimization (SRO)** framework that builds the regression model component by component, as shown in Figure 2. This procedure presents two distinct merits. Firstly, it ensures the orthogonality of the components and provides their ranked importance by imposing geometric constraints on the parameter space, utilizing Riemannian manifold optimization techniques. Secondly, it decomposes the global optimization problem into smaller subproblems, effectively reducing the difficulty of the optimization process.

At each stage $k$, the new basis matrix $\mathbf{B}_{i,p}$ is optimized under strict orthonormality within the component ($\mathbf{B}_{i,p_i}^\top \mathbf{B}_{i,p_i} = \mathbf{I}_{b_i}$) and orthogonality to all previous components ($\mathbf{B}_{i,p_i}^\top \mathbf{B}_{i,q} = \mathbf{0}_{b_i \times b_i}, \ \forall q < p_i$). This sequential, constrained optimization ensures that each new component must discover predictive patterns in a subspace orthogonal to those already captured. Consequently, the extraction order naturally reflects the components' importance in reducing the overall prediction error.

A purely greedy sequential approach, where past decisions are immutable, often leads to suboptimal global solutions. To mitigate this, OSCAR incorporates a **Cooperative Refinement Mechanism**. **Synergy Matrix** $\mathbf{A}_{i,k} \in \mathbb{R}^{b_i \times r_i}$ projects the $b_i$-dimensional representation learned by $\mathbf{B}_{i,p_i}$ into $r_i$ ($r_i < b_i$) dimensions, capturing information that is most useful for cross-mode synergies.

We now reformulate the model structure. The regression tensor can be expressed as:

$$\mathcal{B} = \sum_{\mathbf{k}=1}^{K} \mathcal{W}_k \times_1 \mathbf{M}_{1,k} \times_2 \mathbf{M}_{2,k} \cdots \times_n \mathbf{M}_{n,k}, \tag{10}$$

where $\mathbf{M}_{i,k} = \mathbf{B}_{i,p_i} \mathbf{A}_{i,k} \in \mathbb{R}^{d_i \times r_i}$ denotes the **Effective Projection Operator** for component $k$ with combination $\mathbf{p}$ on mode $i$. When learning the $k$-th principal component with combination $\mathbf{p}$, the algorithm does not merely optimize the new parameters $\{\mathbf{B}_{i,p_i}, \mathbf{A}_{i,k}\}_{i=1}^n$ and the regression head $\mathcal{W}$. Crucially, it also simultaneously re-optimizes **all previously learned synergy matrices**, $\{\mathbf{A}_{i,q}\}_{q=1}^{k-1}$ for all modes $i$. This is possible because at stage $k$, the feature tensor

**Algorithm 1** OSCAR-IMOBT (Joint Training)

---

**Require:** Training tensors $\{\mathcal{X}_s, \mathcal{Y}_s\}_{s=1}^N$; interaction tuples $\mathcal{P}$; mode-wise dims $(d_i, b_i)$; components index set $\mathcal{K}$; learning rates; regularization $\lambda$.

1: Initialize all $\mathbf{B}_{i,p_i} \in \mathrm{St}(d_i, b_i)$, and regression head $\mathcal{W}$.

2: best_loss $\leftarrow +\infty$.

3: **repeat**

4:     **Core construction** (using IMOBT, Eqs. 7, 8, 9):

$$\mathcal{X}_{\mathrm{core}}^{(k)} = \mathcal{X} \times_1 \mathbf{B}_{1,p_1}^\top \cdots \times_n \mathbf{B}_{n,p_n}^\top,$$

    and stack $\{\mathcal{X}_{\mathrm{core}}^{(k)}\}_{k \in \mathcal{K}}$ into $\mathcal{X}_c$.

5:     **Update regression head** $\mathcal{W}$ by ridge regression on $(\mathcal{X}_c, \mathcal{Y})$ (closed-form solution; see also Appendix if desired).

6:     **Update feature bases** $\{\mathbf{B}_{i,p_i}\}$: for each $i, p$, perform a Riemannian gradient step on $\mathrm{St}(d_i, b_i)$ using *Algorithm 4 (Appendix)*.

7:     Compute current training loss and update best_loss if improved.

8: **until** convergence or early stopping

9: **return** $\{\mathbf{B}_{i,p_i}\}$ and regression head $\mathcal{W}$.

---

**Algorithm 2** OSCAR-SRO (Sequential with Cooperative Refinement)

---

**Require:** Same inputs and hyperparameters as Algorithm 1 and mode-wise dims $(d_i, b_i, r_i)$.

1: Same initialization as Algorithm 1 and synergy matrices $A_{i,k} \in \mathrm{St}(b_i, r_i)$.

2: **for** $t = 1$ to $t = max(P_1, ...P_n)$ **do**

3:     **repeat**

4:         Select combination tuple $\mathbf{p} = (p_1, \ldots, p_n) \in \mathcal{P}$.

5:         Set $\mathbf{M}_{i,p_i} = \mathbf{B}_{i,p_i} A_{i,k}$ for all modes $i$.

6:         Build $\mathcal{X}_c$ using Eqs. 7, 8 and 10,

7:         **Update regression head** $W$ by ridge regression on the current $\mathcal{X}_c$.

8:         **Cooperative refinement of** $\{\mathbf{A}_{i,k}\}$: using *Algorithm 3 (Appendix)*.

9:         **Subspace-constrained update of new basis** $\mathbf{B}_{i,k}$: for each mode $i$, update $\mathbf{B}_{i,k}$ using *Algorithm 5 (Appendix)*.

10:     **until** convergence or early stopping

11: **end for**

12: **return** $\{\mathbf{B}_{i,p_i}\}$, $\{\mathbf{A}_{i,k}\}$, effective projections $\mathbf{M}_{i,p} = \mathbf{B}_{i,p_i} \mathbf{A}_{i,p}$, and regression head $\mathcal{W}$.

---

$\mathcal{X}_c$ is constructed using all components up to $k$. Consequently, the loss gradient naturally flows back to all participating synergy matrices during backpropagation.

This look-back refinement allows the established components (whose bases $\mathbf{B}_{i,p_i}$ are now fixed and orthogonal) to recalibrate their contributions and interactions in light of the new information brought by the $k$-th component. It transforms a rigid, greedy sequence into a dynamic, collaborative process, where all components work in concert to achieve a superior global solution. This mechanism endows our sequential learning process with an iterative global consideration, aiming to compensate for the short-sightedness of traditional deflation methods and significantly enhancing the model's overall predictive performance.

We now formalize the whole optimization framework. At each stage $t$ of our sequential framework, the optimization problem can be stated as:

$$\min_{\Theta} \mathcal{L}(\Theta) = \frac{1}{N} \sum_{s=1}^{N} ||\mathcal{Y}_s - \langle \mathcal{B}, \mathcal{X}_s \rangle_{\{1, \cdots, n\}}||_F^2 + \sum_{k=1}^{t} \lambda ||\mathcal{W}_k||^2 \quad \text{subject to} \quad \mathbf{B}_{i,p_i} \in \mathcal{M}_{i,p_i}, \ \mathbf{A}_{i,k} \in \mathcal{N}_i \qquad (11)$$

where $N$ is the sample size, $\lambda$ is the regularization parameter. If the $t$th component has combination $\mathbf{p} = (p_1, \cdots, p_n)$, then $\Theta = \{\mathbf{B}_{i,p_i}, \mathbf{A}_{i,k}, \mathcal{W}_k, k \leq t\} / \{\mathbf{B}_{i,p_i} \text{ that appear in previous stages}\}$. $\mathcal{L}(\Theta)$ is the overall prediction loss. $\mathcal{B}$ is defined in Eq. 10. $\mathcal{M}_{i,p_i}$ and $\mathcal{N}_i$ are special Stiefel manifolds defined by the constraints:

$$\mathcal{M}_{i,p_i} = \{\mathbf{B} \in \mathbb{R}^{d_i \times b_i} \mid \mathbf{B}^\top \mathbf{B} = \mathbf{I}, \mathbf{B}^\top \mathbf{Q}_{<p_i} = \mathbf{0}\},$$

$$\mathcal{N}_i = \{\mathbf{A} \in \mathbb{R}^{b_i \times r_i} \mid \mathbf{A}^\top \mathbf{A} = \mathbf{I}\}.$$

Here, $\mathbf{Q}_{<p_i} = [\mathbf{B}_{i,1}, \ldots, \mathbf{B}_{i,p_i-1}]$ represents the subspace spanned by all previously learned components in mode $i$. We sequentially solve the problem in Eq. 11 $K$ times to obtain $K$ components. To solve the problem in Eq. 11, we develop a **Subspace Constrained Riemannian Gradient Descent (SCRGD)** algorithm as detailed in Appendix C. This method computes the gradient in the ambient Euclidean space, projects it onto the tangent space of the manifold $\mathcal{M}_{i,p}$, and then uses a retraction operation to ensure the updated matrix strictly remains on the manifold. This principled geometric approach guarantees that all orthogonality constraints are rigorously maintained throughout training. The full derivation and algorithmic details are provided in Appendix D.

## 4 EXPERIMENTS

### 4.1 STUDY 1: EFFICACY OF THE IMOBT STRUCTURE

#### 4.1.1 OBJECTIVE AND SETUP FOR JOINT TRAINING

*A. Synthetic Data*

To validate our model under different conditions, we generated various synthetic datasets that conform to the population model. We first construct a set of "true" parameters and a noise distribution, then sample input tensors $\mathcal{X}$ and compute the corresponding responses $\mathcal{Y}$ according to this model. Details are provided in Appendix E.1. For this simulation, we set the number of samples $N = 2000$, the input tensor dimensions to $(50, 50, 10)$, and the output tensor dimensions to $(5, 5)$. Additive Gaussian noise $\mathcal{E} \sim \mathcal{N}(0, 0.1\mathbf{I})$ was added to the response. We conducted ablation studies on the component number $K$, the size of the feature extraction matrix $\{\mathbf{B}_i\}$, and the order of input tensor $\mathcal{X}$, with the results presented in Figure 3. We evaluate predictive accuracy using the coefficient of determination $R^2 = 1 - \sum_{i=1}^{N}(y_i - \hat{y}_i)^2 / \sum_{i=1}^{N}(y_i - \bar{y})^2$, where $\bar{y}$ is the mean of $\{y_i\}$.

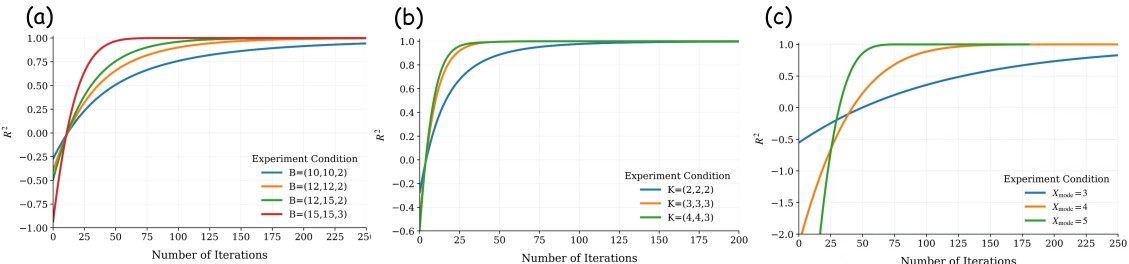

Figure 3: Results of ablation studies on the sizes of the feature extraction matrices, the number of components $K$ and the order of $\mathcal{X}$ on synthetic data. For the first two scenarios, the input is a third-order tensor. $B = (b_1, b_2, b_3)$ denotes the projection dimensions of the feature extraction matrices along the three tensor modes, and $K = (k_1, k_2, k_3)$ denotes the number of feature extraction matrices used in each mode.

*B. Real Data*

We evaluate our proposed method on two publicly available real-world datasets: the Labeled Faces in the Wild (LFW) dataset and an Electrocorticography (ECoG) dataset from a macaque monkey.

**LFW Dataset.** The LFW dataset (Kawulok et al., 2016) is a large collection of face images. Following the tensor-on-tensor regression setup in Lock (2018); Qin & Zhu (2025), our goal is to predict a 73-dimensional facial attribute vector $\mathbf{y}_i \in \mathbb{R}^{73}$ from a corresponding face image tensor $\mathcal{X}_i \in \mathbb{R}^{H \times W \times C}$ (Height × Width × Channel). We first filter the dataset to ensure all images are readable, resulting in 13,143 usable samples. We then create a fixed data split by randomly selecting 10,000 samples for the training set and using the remaining 3,143 for the test set, with the random state seeded for reproducibility.

**ECoG Monkey Dataset.** This dataset, first introduced in Shimoda et al. (2012), involves predicting the 3D hand position of a monkey from its ECoG brain signals. While it has been utilized in a Tensor-on-Scalar regression context (Zhou et al., 2024), our work retains the original problem structure by framing it as a Tensor-on-Tensor task. Following a comprehensive preprocessing pipeline, each sample's predictor is a tensor $\mathcal{X}_i \in \mathbb{R}^{64 \times 10 \times 10}$, representing the normalized spectral power across Channels, Frequencies, and historical time Lags. The corresponding response is a vector $\mathbf{y}_i \in \mathbb{R}^3$, representing the 3D coordinates of the monkey's wrist. The cleaned dataset is split chronologically: the first 10 minutes of data are used for training, and the subsequent 5 minutes serve as the test set. The full details of the preprocessing pipeline, including filtering and feature extraction, are provided in Appendix E.2.

**Noisy Label Generation.** To assess model robustness, we simulate noisy label conditions by corrupting the response vectors. Crucially, noise is added *exclusively to the training set*, while the test set remains clean for fair evaluation. For each training sample $(\mathcal{X}_i^{\text{train}}, \mathbf{y}_i^{\text{train}})$, we generate a noisy label $\tilde{\mathbf{y}}_i^{\text{train}}$ as follows:

$$\tilde{\mathbf{y}}_i^{\text{train}} = \mathbf{y}_i^{\text{train}} + \boldsymbol{\epsilon}_i, \quad \text{where} \quad \boldsymbol{\epsilon}_i \sim \mathcal{N}(0, \gamma^2 \mathbf{I})$$

We experiment with three levels of noise variance: for ECoG, $\gamma^2 \in \{1, 3, 5\}$; for LFW, $\gamma^2 \in \{0.03, 0.05, 0.1\}$.

Table 1: Comparison of methods on real datasets with varying $\gamma^2$.

| Method | ECoG | | | | | | | | LFW | | | | | | | |
|---|---|---|---|---|---|---|---|---|---|---|---|---|---|---|---|---|
| | Real | | Noise Case1 | | Noise Case2 | | Noise Case3 | | Real | | Noise Case1 | | Noise Case2 | | Noise Case3 | |
| | $r \uparrow$ | RPE $\downarrow$ | $r \uparrow$ | RPE $\downarrow$ | $r \uparrow$ | RPE $\downarrow$ | $r \uparrow$ | RPE $\downarrow$ | $r \uparrow$ | RPE $\downarrow$ | $r \uparrow$ | RPE $\downarrow$ | $r \uparrow$ | RPE $\downarrow$ | $r \uparrow$ | RPE $\downarrow$ |
| FullReg | 0.9670 | 0.2392 | 0.9670 | 0.2392 | 0.9670 | 0.2393 | 0.9670 | 0.2390 | 0.5754 | 1.1214 | 0.5538 | 1.1783 | 0.5442 | 1.2122 | 0.5169 | 1.3030 |
| CPReg (Lock, 2018) | 0.9795 | 0.1895 | 0.9813 | 0.1806 | 0.9794 | 0.1905 | 0.9775 | 0.1998 | 0.8283 | 0.5399 | 0.8258 | 0.5433 | 0.8279 | 0.5404 | 0.8298 | 0.5378 |
| TuckerReg (Gahrooei et al., 2021) | 0.9749 | 0.2107 | 0.9781 | 0.1997 | 0.9795 | 0.1907 | 0.9777 | 0.2019 | 0.7963 | 0.5828 | 0.7971 | 0.5819 | 0.7963 | 0.5829 | 0.7921 | 0.5882 |
| TTReg(IHT)(Qin & Zhu, 2025) | 0.9809 | 0.1827 | 0.9810 | 0.1826 | 0.9810 | 0.1826 | 0.9810 | 0.1826 | 0.8231 | 0.5472 | 0.8230 | 0.5473 | 0.8230 | 0.5474 | 0.8237 | 0.5465 |
| TTReg(RGD) (Qin & Zhu, 2025) | 0.9816 | 0.1801 | 0.9815 | 0.1802 | 0.9814 | 0.1803 | 0.9814 | 0.1811 | 0.7343 | 0.6542 | 0.7240 | 0.6647 | 0.7258 | 0.6631 | 0.7132 | 0.6754 |
| OSCAR-IMOBT (ours) | **0.9822** | **0.1761** | **0.9816** | **0.1793** | **0.9824** | **0.1751** | **0.9825** | **0.1745** | **0.8446** | **0.5159** | **0.8406** | **0.5219** | **0.8386** | **0.5249** | **0.8339** | **0.5317** |

**Baselines and Evaluation Metrics.** We compare our model against the following regression methods: FullReg (unregularized), CPReg (Lock, 2018), TuckerReg (Gahrooei et al., 2021), and two variants of TTReg from (Qin & Zhu, 2025), namely TTReg(IHT) and TTReg(RGD). Performance is evaluated using two metrics: The Pearson correlation coefficient ($r$) between the predicted and true responses, averaged across all response dimensions, and the Relative Prediction Error (RPE), defined as:

$$\text{RPE} = \frac{\|\mathbf{Y}_{\text{test}} - \hat{\mathbf{Y}}_{\text{test}}\|_F^2}{\|\mathbf{Y}_{\text{test}}\|_F^2}$$

where $\mathbf{Y}_{\text{test}}$ and $\hat{\mathbf{Y}}_{\text{test}}$ are the ground-truth and predicted response matrices for the test set, respectively, and $\|\cdot\|_F$ denotes the Frobenius norm. To provide a more intuitive scale for comparison, the RPE values reported in the subsequent tables are the square root of this value.

### 4.1.2 PERFORMANCE ANALYSIS

Table 1 summarizes the predictive performance of OSCAR-IMOBT (the jointly trained version of our model) against state-of-the-art baselines on the ECoG and LFW datasets. On both datasets, OSCAR-IMOBT achieves the best performance across all metrics, registering the highest correlation ($r$) and the lowest Relative Prediction Error (RPE). For instance, on the clean ECoG dataset, our method attains an RPE of **0.1761**, outperforming the next best method, TTReg(RGD), which scored 0.1801. A similar trend is observed on the LFW dataset. Crucially, our model demonstrates superior robustness in the presence of label noise. As the noise variance ($\gamma^2$) increases in the training data, the performance of baseline methods tends to degrade more significantly. In contrast, OSCAR-IMOBT maintains a stable and strong performance, with the performance gap often widening under noisy conditions. For example, in LFW Noise Case 3, our model's RPE is **0.5317**, whereas the next best competitor, CPReg, is at 0.5378. This resilience can be attributed to the strong regularization effect induced by the orthogonality constraints on the feature extraction matrices $\mathbf{B}_{i,p_i}$, which prevents the model from overfitting to the noise in the labels.

### 4.2 STUDY 2: EFFICACY OF THE OSCAR-SRO FRAMEWORK

### 4.2.1 ANALYSIS OF COMPONENT PREDICTIVE POWER

To directly evaluate the quality and ordering of the principal components extracted by our framework, we conducted a comparative experiment against HOPLS, a representative sequential component extraction method based on data-space deflation. We configured both OSCAR-SRO and HOPLS to sequentially extract the same number of principal components ($K = 4$) with identical dimensionality, using the real-world ECoG dataset setup. To isolate the predictive ability of each component, we evaluated them separately: for each component, we trained a new $L_2$-regularized FC layer using only that component as input, and recorded the test-set RPE independently. The results are illustrated in Figure 4. The RPE curve for OSCAR-SRO exhibits a significantly steeper initial descent compared to HOPLS. Furthermore, a particularly noteworthy finding is the superior quality of OSCAR's initial component. As shown at $K = 1$, the RPE achieved by OSCAR's single, refined component is substantially lower than that of HOPLS's first component. This highlights the power of our Cooperative Refinement Mechanism. The learnable synergy matrix $\mathbf{A}_{i,1}$ is not just optimized in the first stage; it is re-optimized and refined as subsequent components are added. This "look-back" capability ensures that the final version of the first component is not merely a product of a single greedy step but is a sophisticated feature basis that has been tuned for optimal synergy with the broader component set. This leads to a performance level for the first component that is difficult for a model trained in a single, isolated stage to achieve. In summary, this analysis provides strong evidence that OSCAR-SRO excels not only in overall predictive accuracy but also in discovering a more efficiently ordered and synergistically refined set of principal components.

Figure 4: Predictive performance of individual components extracted by OSCAR-SRO and HOPLS on the ECoG dataset. Each component is evaluated using an $L_2$-regularized FC layer.

### 4.2.2 ABLATION STUDY OF OSCAR

Table 2 presents an ablation study on the ECoG dataset to dissect the contributions of the SRO framework and the Cooperative Refinement Mechanism. In the table, 'Diag X' refers to jointly trained IMOBT models with different settings, while "Diag0.5+Sequential" represents our full OSCAR model using SRO.

The key finding is that the sequential model ("Diag0.5+Sequential") consistently achieves the best performance across all noise conditions, obtaining the lowest RPE. For instance, on the clean data, it achieves an RPE of **0.1721**, surpassing the best jointly trained model ("Diag 0.5") which scored 0.1730. This result empirically validates our central hypothesis: the SRO framework, by replacing data-space deflation with parameter-space geometric constraints and incorporating a look-back refinement step, overcomes the greedy sub-optimality of traditional sequential methods and converges to a higher-quality solution. The synergy matrix $\mathbf{A}$, re-optimized at each stage, allows existing components to adapt to new ones, fostering a collaborative rather than a purely greedy search for the optimal regression tensor. This demonstrates the significant practical benefit of our proposed sequential learning paradigm.

Table 2: Ablation Study Results in ECoG

|  | Diag Full | Diag 0.5 | Diag 1 | Diag 1.5 | Diag 2 | Diag 0.5 + Sequential |
|---|---|---|---|---|---|---|
| Real | 0.1810(0.9812) | 0.1730(0.9828) | 0.1757(0.9823) | 0.1761(0.9822) | 0.1771(0.9820) | **0.1721**(0.9830) |
| Noise Case 1 | 0.1811(0.9812) | 0.1736(0.9827) | 0.1757(0.9823) | 0.1780(0.9818) | 0.1766(0.9821) | **0.1712**(0.9832) |
| Noise Case 2 | 0.1849(0.9804) | 0.1734(0.9828) | 0.1767(0.9821) | 0.1757(0.9823) | 0.1753(0.9824) | **0.1727**(0.9829) |
| Noise Case 3 | 0.1851(0.9795) | 0.1730(0.9828) | 0.1759(0.9823) | 0.1776(0.9819) | 0.1756(0.9823) | **0.1730**(0.9828) |

## 5 CONCLUSION

In this work, we addressed the key challenges in tensor-on-tensor regression by proposing OSCAR, a novel framework that unifies low-rank modeling with supervised, orthogonal component extraction. OSCAR introduces a paradigm shift from classical data-space deflation by employing a Sequential Riemannian Optimization (SRO) framework, which enforces component orthogonality directly in the parameter space to prevent error propagation. This is enhanced by a Cooperative Refinement Mechanism that mitigates the greedy bias of sequential learning, enabling a "look-back" capability for a superior global solution. Our Input-Mode Orthogonal Block Term (IMOBT) structure provides a flexible, generalized view of Block Term decomposition and Tucker decomposition, enabling the model to achieve superior results with fewer parameters. Extensive experiments demonstrated OSCAR's efficacy, achieving state-of-the-art predictive performance on real-world datasets, particularly under noisy conditions. Our analysis confirmed that OSCAR not only excels in accuracy but also discovers a more efficiently ordered and synergistically refined set of principal components than traditional methods. In conclusion, OSCAR provides a robust and interpretable framework for tensor-on-tensor regression.

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

## A  NOTATION

The main notations used in this paper are summarized in Table 3.

## B  CONVERGENCE OF POPULATION MODEL

### B.1  NOTATION AND PRELIMINARIES

For self-containment, we restate here the notation used in our theoretical analysis.

- **Data tensors.** The input is $\mathcal{X} \in \mathbb{R}^{N \times d_1 \times \cdots \times d_n}$, and the response is $\mathcal{Y} \in \mathbb{R}^{N \times o_1 \times \cdots \times o_q}$. For sample $s \in [N] := \{1, \ldots, N\}$, we denote

$$\mathcal{X}_s \in \mathbb{R}^{d_1 \times \cdots \times d_n}, \qquad \mathcal{Y}_s \in \mathbb{R}^{o_1 \times \cdots \times o_q}.$$

- **Vectorization and inner products.** The operator $\text{vec}(\cdot)$ stacks all entries of a tensor into a column vector. The Frobenius inner product between tensors $\mathcal{A}$ and $\mathcal{B}$ of the same size is

$$\langle \mathcal{A}, \mathcal{B} \rangle := \sum_i \text{vec}(\mathcal{A})_i \, \text{vec}(\mathcal{B})_i,$$

  with corresponding Frobenius norm $\|\mathcal{A}\|_F = \sqrt{\langle \mathcal{A}, \mathcal{A} \rangle}$.

- **i-mode product and Kronecker product.** For a tensor $\mathcal{X}$ and a matrix $\mathbf{U}$, the notation $\mathcal{X} \times_i \mathbf{U}$ denotes the mode-$i$ tensor–matrix product. The Kronecker product of matrices is denoted by $\otimes$.

- **Per-mode feature operators.** For a component index (interaction tuple) $p = (p_1, \ldots, p_n) \in \mathcal{P}$ and mode $i \in [n]$, we define

$$\mathbf{B}_{i,p_i} \in \mathbb{R}^{d_i \times b_i}, \qquad \mathbf{A}_{i,k} \in \mathbb{R}^{b_i \times r_i}, \qquad \mathbf{M}_{i,p_i} := \mathbf{B}_{i,p_i} \mathbf{A}_{i,k} \in \mathbb{R}^{d_i \times r_i}.$$

  Columns of $\mathbf{B}_{i,p_i}$ form a basis, and $\mathbf{A}_{i,k}$ mixes these basis features to form $r_i$ latent coordinates. The matrices $\mathbf{M}_{i,p_i}$ are the *effective projection operators* on each mode. Note that different tuples $\mathbf{p}, \mathbf{p}' \in \mathcal{P}$ may share the same per-mode index $p_i$, so the same basis matrix $\mathbf{B}_{i,p_i}$ can be reused across components.

- **Combination set (interaction scheme).** A tuple $\mathbf{p} = (p_1, \ldots, p_n) \in \mathcal{P}$ selects one effective operator $\mathbf{M}_{i,p_i}$ per mode. Canonical choices include

$$\mathcal{P}_{\mathrm{diag}} = \{(p, \ldots, p) \mid p = 1, \ldots, P\}, \qquad \mathcal{P}_{\mathrm{full}} = \{1, \ldots, P\}^n.$$

- **Per-sample feature vectorization.** For a sample $s$ and the $k$-th component associated with tuple $\mathbf{p} = (p_1, \ldots, p_n)$, the core feature tensor is

$$\mathcal{X}_{\mathrm{core},s}^{(k)} := \mathcal{X}_s \times_1 \mathbf{M}_{1,p_1}^\top \cdots \times_n \mathbf{M}_{n,p_n}^\top \in \mathbb{R}^{r_1 \times \cdots \times r_n}.$$

  Its vectorized form is

$$z_{\mathrm{core},s}^{(k)} := \mathrm{vec}\big(\mathcal{X}_{\mathrm{core},s}^{(k)}\big) \in \mathbb{R}^R, \qquad R := \prod_{i=1}^n r_i.$$

  Stacking all component-wise feature vectors $\{z_{\mathrm{core},s}^{(k)}\}_{k=1}^K$ yields a single feature vector

$$z_s \in \mathbb{R}^D, \qquad D := K\,R.$$

  We index components by $k \in [K] := \{1, \ldots, K\}$ and fix a bijection between component indices and tuples, denoted $\mathbf{p}(k) \in \mathcal{P}$. For notational convenience, we also write

$$z_{\mathbf{p},s} := z_{\mathrm{core},s}^{(k)} \quad \text{whenever} \quad \mathbf{p} = \mathbf{p}(k),$$

  and use $z_{\mathbf{p}}(\cdot)$ to denote the corresponding feature map $\mathcal{X} \mapsto z_{\mathbf{p}}(\mathcal{X})$.

- **Regression head.** Let the flattened output dimension be $m_{\mathrm{out}} := \prod_{j=1}^m o_j$. Using a weight matrix $\mathbf{W} \in \mathbb{R}^{D \times m_{\mathrm{out}}}$, the prediction for sample $s$ is

$$\widehat{\mathbf{y}}_s = \mathbf{W}^\top z_s, \qquad \mathbf{y}_s := \mathrm{vec}\big(\mathcal{Y}_s\big) \in \mathbb{R}^{m_{\mathrm{out}}}.$$

  In the main text, we use the tensor $\mathcal{W}_k$ to represent the coefficient tensor of the prediction head. However, this is equivalent to vectorizing the core tensor into $\mathbf{z}_s$, and then using a matrix $\mathbf{W}_k$ to obtain $\mathbf{y}_s$. For conciseness, we will adopt this approach to represent the prediction head in the following sections.

## B.2 STANDING ASSUMPTIONS

**Assumption 1** (Separable Tensor-Variate Normal (TVN) Design)**.** *For each sample $s$, the vectorized input follows*

$$\mathrm{vec}\big(\mathcal{X}_s\big) \sim \mathcal{N}\big(0,\ \mathbf{\Sigma}_n \otimes \cdots \otimes \mathbf{\Sigma}_1\big),$$

*where each $\mathbf{\Sigma}_i \in \mathbb{R}^{d_i \times d_i}$ is symmetric positive definite. The errors $\varepsilon_s$ are independent across samples and satisfy $\mathbb{E}[\varepsilon_s \mid \mathcal{X}_s] = 0$.*

**Assumption 2** (Global $\mathbf{\Sigma}_i$-Orthogonality Constraints (Generalized Stiefel))**.** *We restrict the parameter space for $(\mathbf{B}, \mathbf{A})$ to the generalized Stiefel manifolds defined as follows. For each mode $i \in [n]$ and each per-mode index $p_i$, the matrices $\mathbf{B}_{i,p_i}$ and $\mathbf{A}_{i,k}$ are constrained to satisfy*

$$\mathbf{B}_{i,p_i}^\top \mathbf{\Sigma}_i \mathbf{B}_{i,p_i} = \mathbf{I}_{b_i}, \qquad \mathbf{A}_{i,k}^\top \mathbf{A}_{i,k} = \mathbf{I}_{r_i},$$

*and for $p_i \neq p_i'$,*

$$\mathbf{B}_{i,p_i}^\top \mathbf{\Sigma}_i \mathbf{B}_{i,p_i'} = 0.$$

*Consequently, the effective projection matrices satisfy*

$$\mathbf{M}_{i,p_i}^\top \mathbf{\Sigma}_i \mathbf{M}_{i,p_i} = \mathbf{I}_{r_i}, \qquad \mathbf{M}_{i,p_i}^\top \mathbf{\Sigma}_i \mathbf{M}_{i,p_i'} = 0 \quad \text{for } p_i \neq p_i'.$$

*The ground-truth parameters $(\mathbf{B}^\star, \mathbf{A}^\star)$ lie in this constrained parameter space.*

**Assumption 3** (Component-level Discriminability). *Let $\mathcal{P}^\star \subseteq \mathcal{P}$ be the set of true component tuples. For any distinct $\mathbf{p}, \mathbf{p}' \in \mathcal{P}^\star$, there exists at least one mode $i$ such that*

$$\mathbf{M}_{i,p_i}^{\star\top} \mathbf{\Sigma}_i \mathbf{M}_{i,p_i'}^\star = 0.$$

**Assumption 4** (Signal Non-degeneracy). *Let $\mathbf{W}^\star = [\mathbf{W}_k^\star]_{k\in\mathcal{K}^\star} \in \mathbb{R}^{D \times m_{\mathrm{out}}}$ be the true weight matrix, where each $\mathbf{W}_k^\star$ is the block of rows associated with component tuple $\mathbf{p}$. There exists a constant $\alpha > 0$ such that*

$$\min_{k\in\mathcal{K}^\star} \|\mathbf{W}_k^\star\|_F \ \geq \ \alpha.$$

We also define the index set of *true* components

$$K^\star := \{k \in [K] : \mathbf{p}(k) \in \mathcal{P}^\star\},$$

which is the collection of component indices corresponding to the true tuple set $\mathcal{P}^\star$.

**Assumption 5** (Identifiability up to Natural Equivalences). *Any two parameter tuples $(\mathbf{B}, \mathbf{A})$ and $(\bar{\mathbf{B}}, \bar{\mathbf{A}})$ that realize the same regression function $S(\mathcal{X}) := \mathbb{E}[\boldsymbol{y} \mid \mathcal{X}]$ are equivalent up to: (i) permutation of the components in $\mathcal{P}^\star$; and (ii) per-mode, within-subspace orthogonal rotations of the bases $\{\mathbf{M}_{i,p_i}^\star\}$. In particular, the ground-truth parameters are identified up to these natural equivalences.*

**Remark 1** (Whitening Equivalence). *Under Assumption 1, we can define a whitened tensor*

$$\widetilde{\mathcal{X}} = \mathcal{X} \times_1 \mathbf{\Sigma}_1^{-1/2} \cdots \times_n \mathbf{\Sigma}_n^{-1/2}$$

*and whitened parameters $\widetilde{\mathbf{M}}_{i,p_i} = \mathbf{\Sigma}_i^{1/2} \mathbf{M}_{i,p_i}$. Then $\mathrm{vec}(\widetilde{\mathcal{X}}) \sim \mathcal{N}(0, \mathbf{I})$ and all $\mathbf{\Sigma}_i$-orthogonality conditions reduce to standard Euclidean orthogonality. The proofs below are carried out in this whitened coordinate system without loss of generality.*

## B.3 MAIN RESULT

**Theorem 1** (Population Optimality, Local Geometry, and Convergence). *Suppose Assumptions 1–5 hold. Then:*

(I) ***Population optimality of the regression function.*** *For any feasible $(\mathbf{B}, \mathbf{A})$ satisfying the global orthogonality constraints, the population risk*

$$\mathcal{L}(\mathbf{B}, \mathbf{A}, \mathbf{W}) = \frac{1}{2}\,\mathbb{E}\big[\|\mathbf{W}^\top z - \boldsymbol{y}\|_2^2\big] + \frac{\lambda}{2}\|\mathbf{W}\|_F^2$$

*admits the reduced form*

$$\widetilde{\mathcal{L}}(\mathbf{B}, \mathbf{A}) := \min_{\mathbf{W}} \mathcal{L}(\mathbf{B}, \mathbf{A}, \mathbf{W}) = \mathrm{const} - \frac{1}{2(1+\lambda)}\|\mathbf{\Sigma}_{z\boldsymbol{y}}\|_F^2 = \mathrm{const} - \frac{1}{2(1+\lambda)}\|\Pi_{\mathcal{H}(\mathbf{B},\mathbf{A})}S\|_{L^2}^2,$$

*where $\mathbf{\Sigma}_{z\boldsymbol{y}} = \mathbb{E}[z\boldsymbol{y}^\top]$, $S(\mathcal{X}) := \mathbb{E}[\boldsymbol{y} \mid \mathcal{X}]$, and $\Pi_{\mathcal{H}(\mathbf{B},\mathbf{A})}$ is the $L^2$-orthogonal projector onto the feature space $\mathcal{H}(\mathbf{B}, \mathbf{A}) := \mathrm{span}\{z_p(\cdot) : \mathbf{p} \in \mathcal{P}\}$. As a consequence, population risk is minimized if and only if $S \in \mathcal{H}(\mathbf{B}, \mathbf{A})$; in particular, at the ground-truth parameters $(\mathbf{B}^\star, \mathbf{A}^\star)$ the model recovers the true regression function exactly. Under Assumption 5, any two minimizers represent the same regression function if and only if they are equivalent up to permutation and per-mode orthogonal rotations.*

(II) ***Local Riemannian geometry.*** *In a neighborhood of the true equivalence class of $(\mathbf{B}^\star, \mathbf{A}^\star)$, the reduced risk $\widetilde{\mathcal{L}}(\mathbf{B}, \mathbf{A})$ is Riemannian $\mu$-strongly convex and $L$-smooth on the normal tangent space of the product Stiefel manifold, with*

$$\mu = \frac{1}{1+\lambda} \min_{i\in[n]} \Big(\alpha^2 \prod_{j\neq i} r_j\Big) \ > \ 0.$$

(III) ***Linear convergence of joint Riemannian GD.*** *Consider Riemannian gradient descent applied to the full parameter tuple $\Theta := (\mathbf{B}, \mathbf{A}, \mathbf{W})$ on the product manifold formed by the Stiefel constraints for $(\mathbf{B}, \mathbf{A})$ and the Euclidean space for $\mathbf{W}$, with respect to the original population risk $\mathcal{L}(\mathbf{B}, \mathbf{A}, \mathbf{W})$. If initialized within a basin of attraction of the true equivalence class, then there exist constants $\tilde{\mu} > 0$ and $\tilde{L} < \infty$ such that for any stepsize $\eta \in (0, 2/\tilde{L})$ we have linear convergence:*

$$\mathrm{dist}\big(\Theta^{(t)}, \Theta^\star\big) \ \leq \ (1 - \eta\tilde{\mu})^t\, \mathrm{dist}\big(\Theta^{(0)}, \Theta^\star\big),$$

*where $\Theta^\star := (\mathbf{B}^\star, \mathbf{A}^\star, \mathbf{W}^\star)$ and $\mathrm{dist}(\cdot, \cdot)$ denotes the Riemannian distance on the corresponding quotient manifold.*

### B.4 Auxiliary Lemmas

**Lemma 1** (Kronecker Factorization of Features). *For any tuples $\mathbf{p} = (p_1, \ldots, p_n)$ (component $k$) and $\mathbf{p}' = (p'_1, \ldots, p'_n)$ (component $k'$), the core feature vector admits the representation*

$$z_{\text{core}}^{(k)} = \left( \mathbf{M}_{n,p_n}^\top \otimes \cdots \otimes \mathbf{M}_{1,p_1}^\top \right) \text{vec}(\mathcal{X}).$$

*In the whitened coordinate system, this implies*

$$\mathbb{E} \left\langle z_{\text{core}}^{(k)}, z_{\text{core}}^{(k')} \right\rangle = \prod_{i=1}^n \text{tr}\left( \mathbf{M}_{i,p_i}^\top \mathbf{M}_{i,p'_i} \right).$$

*Proof.* The expression for $z_{\text{core}}^{(k)}$ is the standard identity for vectorizing a chain of tensor–matrix products:

$$\text{vec}\left( \mathcal{X} \times_1 \mathbf{M}_{1,p_1}^\top \cdots \times_n \mathbf{M}_{n,p_n}^\top \right) = \left( \mathbf{M}_{n,p_n}^\top \otimes \cdots \otimes \mathbf{M}_{1,p_1}^\top \right) \text{vec}(\mathcal{X}),$$

see Kolda & Bader (2009) for a comprehensive summary of such tensor identities. In the whitened coordinate system, $\mathbb{E}[\text{vec}(\mathcal{X})\text{vec}(\mathcal{X})^\top] = \mathbf{I}$, so

$$\mathbb{E} \left\langle z_{\text{core}}^{(k)}, z_{\text{core}}^{(k')} \right\rangle = \mathbb{E} \, \text{vec}(\mathcal{X})^\top \left( \mathbf{M}_{1,p_1} \otimes \cdots \otimes \mathbf{M}_{n,p_n} \right) \left( \mathbf{M}_{n,p'_n}^\top \otimes \cdots \otimes \mathbf{M}_{1,p'_1}^\top \right) \text{vec}(\mathcal{X})$$

$$= \text{tr}\left( \left( \mathbf{M}_{n,p_n}^\top \mathbf{M}_{n,p'_n} \right) \otimes \cdots \otimes \left( \mathbf{M}_{1,p_1}^\top \mathbf{M}_{1,p'_1} \right) \right)$$

$$= \prod_{i=1}^n \text{tr}\left( \mathbf{M}_{i,p_i}^\top \mathbf{M}_{i,p'_i} \right),$$

where we used $\text{tr}(A \otimes B) = \text{tr}(A)\text{tr}(B)$. $\qquad\square$

**Lemma 2** (Block Isotropy of Features). *Under Assumption 2, the component-wise features are block-isotropic:*

$$\mathbb{E}\left[ z_{\text{core}}^{(k)} (z_{\text{core}}^{(k')})^\top \right] = \delta_{kk'} \, \mathbf{I}_R \qquad \text{for all } k, k' \in \mathcal{K}.$$

*In particular, the full Gram matrix of stacked features is identity, $\boldsymbol{\Sigma}_{zz} := \mathbb{E}[zz^\top] = \mathbf{I}_D$. For the true family $\{z_k^\star\}_{k \in K^\star}$, this property together with Assumption 3 implies that $\{z_k^\star\}$ form an $L^2$-orthonormal system:*

$$\mathbb{E} \left\langle z_k^\star, z_{k'}^\star \right\rangle = \delta_{kk'}, \qquad k, k' \in K^\star.$$

*Proof.* By Lemma 1 and the orthogonality relations in Assumption 2, for $k \neq k'$ there exists at least one mode $i$ for which $\text{tr}(\mathbf{M}_{i,p_i}^\top \mathbf{M}_{i,p'_i}) = 0$, whence

$$\mathbb{E} \left\langle z_{\text{core}}^{(k)}, z_{\text{core}}^{(k')} \right\rangle = 0.$$

For $k = k'$, each $\mathbf{M}_{i,p_i}$ has orthonormal columns, so $\text{tr}(\mathbf{M}_{i,p_i}^\top \mathbf{M}_{i,p_i}) = r_i$ and thus

$$\mathbb{E} \left\langle z_{\text{core}}^{(k)}, z_{\text{core}}^{(k)} \right\rangle = \prod_{i=1}^n r_i = R,$$

which implies that each $z_{\text{core}}^{(k)}$ has identity covariance in $\mathbb{R}^R$:

$$\mathbb{E}\left[ z_{\text{core}}^{(k)} (z_{\text{core}}^{(k)})^\top \right] = \mathbf{I}_R.$$

Stacking over components yields $\boldsymbol{\Sigma}_{zz} = \mathbf{I}_D$. Restricting to the true components $K^\star$ and using Assumption 3 gives the $L^2$-orthonormality of $\{z_k^\star\}_{k \in K^\star}$. $\qquad\square$

**Lemma 3** (Feature Space Factorization). *For each mode $i \in [n]$, define*

$$\mathcal{S}_i := \text{span}\left\{ \text{range}(\mathbf{M}_{i,p_i}) : \mathbf{p} \in \mathcal{P} \right\} \subset \mathbb{R}^{d_i},$$

*where $\text{range}(\mathbf{M})$ denotes the column space of $\mathbf{M}$. Then the feature space*

$$\mathcal{H} := \text{span}\{z_p(\cdot) : \mathbf{p} \in \mathcal{P}\}$$

*coincides with the Hilbert tensor product $\mathcal{H} = \bigotimes_{i=1}^n \mathcal{S}_i$. Similarly, the true feature space is $\mathcal{H}^\star = \bigotimes_{i=1}^n \mathcal{S}_i^\star$, where $\mathcal{S}_i^\star$ is defined analogously using $\mathbf{M}_{i,p_i}^\star$.*

*Proof.* In the whitened coordinates, $\text{vec}(\mathcal{X})$ lives in the finite-dimensional Hilbert space

$$\mathcal{X} := \bigotimes_{i=1}^{n} \mathbb{R}^{d_i},$$

with the natural tensor inner product. For a fixed tuple $p = (p_1, \ldots, p_n)$, Lemma 1 shows that $z_p(\mathcal{X})$ is obtained by applying the product operator

$$\mathbf{T}_p := \mathbf{M}_{n,p_n}^{\top} \otimes \cdots \otimes \mathbf{M}_{1,p_1}^{\top}$$

to $\text{vec}(\mathcal{X})$. Thus the range of $\mathbf{T}_p$ is contained in

$$\text{range}(\mathbf{M}_{n,p_n}^{\top}) \otimes \cdots \otimes \text{range}(\mathbf{M}_{1,p_1}^{\top}) = \text{range}(\mathbf{M}_{n,p_n}) \otimes \cdots \otimes \text{range}(\mathbf{M}_{1,p_1}),$$

where we used the fact that in finite dimensions the range of $\mathbf{M}^{\top}$ is the dual of the range of $\mathbf{M}$, which can be identified with the same subspace.

By definition, $\mathcal{S}_i$ is the span of all such ranges at mode $i$, so the tensor product of the $\mathcal{S}_i$ is contained in $\mathcal{H}$. Conversely, any $z_p$ lies in the algebraic tensor product of the $\mathcal{S}_i$, and finite linear combinations of such tensors are dense in the Hilbert tensor product $\bigotimes_i \mathcal{S}_i$ (Conway, 2019). Therefore

$$\mathcal{H} = \bigotimes_{i=1}^{n} \mathcal{S}_i$$

as Hilbert spaces. The statement for $\mathcal{H}^{\star}$ follows by replacing $\mathbf{M}_{i,p_i}$ with $\mathbf{M}_{i,p_i}^{\star}$ throughout. $\square$

**Lemma 4** (Tensor Product Inclusion). *Let $\mathcal{H} = \bigotimes_{i=1}^{n} \mathcal{S}_i$ and $\mathcal{H}^{\star} = \bigotimes_{i=1}^{n} \mathcal{S}_i^{\star}$ be tensor-product Hilbert spaces. Then $\mathcal{H}^{\star} \subseteq \mathcal{H}$ if and only if $\mathcal{S}_i^{\star} \subseteq \mathcal{S}_i$ for all $i$.*

*Proof.* The "if" direction is immediate: if each $\mathcal{S}_i^{\star} \subseteq \mathcal{S}_i$, then for any pure tensor $u_1^{\star} \otimes \cdots \otimes u_n^{\star}$ with $u_i^{\star} \in \mathcal{S}_i^{\star}$ we also have $u_i^{\star} \in \mathcal{S}_i$, hence $u_1^{\star} \otimes \cdots \otimes u_n^{\star} \in \mathcal{H}$. Since $\mathcal{H}^{\star}$ is the closure of the span of such pure tensors, we obtain $\mathcal{H}^{\star} \subseteq \mathcal{H}$.

For the "only-if" direction, suppose $\mathcal{H}^{\star} \subseteq \mathcal{H}$ but, for contradiction, that $\mathcal{S}_{i_0}^{\star} \nsubseteq \mathcal{S}_{i_0}$ for some $i_0$. Then there exists $v \in \mathcal{S}_{i_0}^{\star} \setminus \mathcal{S}_{i_0}$. Choose nonzero vectors $u_i \in \mathcal{S}_i^{\star}$ for all $i \neq i_0$, and consider the pure tensor

$$u := u_1 \otimes \cdots \otimes u_{i_0-1} \otimes v \otimes u_{i_0+1} \otimes \cdots \otimes u_n \in \mathcal{H}^{\star}.$$

If $\mathcal{H}^{\star} \subseteq \mathcal{H}$ then $u \in \mathcal{H}$. However, by the characterization of tensor-product subspaces (see Kadison & Ringrose (1986)), any pure tensor in $\mathcal{H}$ must have each factor lying in $\mathcal{S}_i$. In particular, the $i_0$-th factor of $u$ must lie in $\mathcal{S}_{i_0}$, which contradicts $v \notin \mathcal{S}_{i_0}$. Therefore $\mathcal{S}_i^{\star} \subseteq \mathcal{S}_i$ for all $i$. $\square$

**Lemma 5** (Projection Energy Identity). *Let $\boldsymbol{S}(\mathcal{X}) := \mathbb{E}[\boldsymbol{y} \mid \mathcal{X}]$ denote the conditional mean (regression function), viewed as an element of $L^2(\Omega, \mathbb{R}^{m_{\text{out}}})$. Under the conditions of Lemma 2, we have*

$$\left\| \boldsymbol{\Sigma}_{z\boldsymbol{y}} \right\|_F^2 = \left\| \Pi_{\mathcal{H}} \boldsymbol{S} \right\|_{L^2}^2,$$

*where $\Pi_{\mathcal{H}}$ is the $L^2$-orthogonal projector onto the feature space $\mathcal{H}$.*

*Proof.* Write $\boldsymbol{S} = (S_1, \ldots, S_{m_{\text{out}}})^{\top}$, where each $S_j$ is scalar-valued. Let $\{z_k\}_{k=1}^{D}$ denote an orthonormal basis of $\mathcal{H}$ in $L^2$, constructed from the component-wise features using Lemma 2. Then for each $j$ we have the expansion

$$\Pi_{\mathcal{H}} S_j = \sum_{k=1}^{D} \langle S_j, z_k \rangle_{L^2} \, z_k,$$

and

$$\| \Pi_{\mathcal{H}} S_j \|_{L^2}^2 = \sum_{k=1}^{D} |\langle S_j, z_k \rangle_{L^2}|^2.$$

By the tower property of conditional expectation and the definition of $S_j$,

$$\langle S_j, z_k \rangle_{L^2} = \mathbb{E}[S_j(\mathcal{X}) z_k(\mathcal{X})] = \mathbb{E}[\mathbb{E}[y_j \mid \mathcal{X}] z_k(\mathcal{X})] = \mathbb{E}[y_j z_k(\mathcal{X})] = [\boldsymbol{\Sigma}_{z\boldsymbol{y}}]_{k,j}.$$

Therefore

$$\| \Pi_{\mathcal{H}} \boldsymbol{S} \|_{L^2}^2 = \sum_{j=1}^{m_{\text{out}}} \| \Pi_{\mathcal{H}} S_j \|_{L^2}^2 = \sum_{j=1}^{m_{\text{out}}} \sum_{k=1}^{D} [\boldsymbol{\Sigma}_{z\boldsymbol{y}}]_{k,j}^2 = \| \boldsymbol{\Sigma}_{z\boldsymbol{y}} \|_F^2.$$

$\square$

### B.5    PROOF OF THEOREM 1

*Proof of Theorem 1.* We prove the three claims in turn.

**(I) Population optimality of the regression function.** By definition,

$$\mathcal{L}(\mathbf{B}, \mathbf{A}, \mathbf{W}) = \frac{1}{2}\mathbb{E}\big[\|\mathbf{W}^\top z - \boldsymbol{y}\|_2^2\big] + \frac{\lambda}{2}\|\mathbf{W}\|_F^2.$$

Let $\boldsymbol{\Sigma}_{zz} := \mathbb{E}[zz^\top]$ and $\boldsymbol{\Sigma}_{z\boldsymbol{y}} := \mathbb{E}[z\boldsymbol{y}^\top]$. Expanding the square and using linearity of expectation,

$$\mathcal{L}(\mathbf{B}, \mathbf{A}, \mathbf{W}) = \frac{1}{2}\mathrm{tr}(\mathbf{W}^\top \boldsymbol{\Sigma}_{zz}\mathbf{W}) - \mathrm{tr}(\mathbf{W}^\top \boldsymbol{\Sigma}_{z\boldsymbol{y}}) + \frac{1}{2}\mathrm{tr}(\boldsymbol{\Sigma}_{\boldsymbol{yy}}) + \frac{\lambda}{2}\mathrm{tr}(\mathbf{W}^\top \mathbf{W}),$$

where $\boldsymbol{\Sigma}_{\boldsymbol{yy}} := \mathbb{E}[\boldsymbol{yy}^\top]$ does not depend on $(\mathbf{B}, \mathbf{A}, \mathbf{W})$. By Lemma 2, for any feasible $(\mathbf{B}, \mathbf{A})$ we have $\boldsymbol{\Sigma}_{zz} = \mathbf{I}_D$, so

$$\mathcal{L}(\mathbf{B}, \mathbf{A}, \mathbf{W}) = \frac{1}{2}\mathrm{tr}\big(\mathbf{W}^\top(\mathbf{I} + \lambda\mathbf{I})\mathbf{W}\big) - \mathrm{tr}(\mathbf{W}^\top \boldsymbol{\Sigma}_{z\boldsymbol{y}}) + \mathrm{const}.$$

The objective is strictly convex in $\mathbf{W}$ and its unique minimizer satisfies

$$(\mathbf{I} + \lambda\mathbf{I})\mathbf{W}^\dagger - \boldsymbol{\Sigma}_{z\boldsymbol{y}} = 0,$$

that is,

$$\mathbf{W}^\dagger(\mathbf{B}, \mathbf{A}) = (1 + \lambda)^{-1}\boldsymbol{\Sigma}_{z\boldsymbol{y}}.$$

Substituting $\mathbf{W}^\dagger$ back into $\mathcal{L}$ yields

$$\widetilde{\mathcal{L}}(\mathbf{B}, \mathbf{A}) := \min_{\mathbf{W}}\mathcal{L}(\mathbf{B}, \mathbf{A}, \mathbf{W}) = \mathrm{const} - \frac{1}{2(1 + \lambda)}\|\boldsymbol{\Sigma}_{z\boldsymbol{y}}\|_F^2.$$

By Lemma 5, this can be rewritten as

$$\widetilde{\mathcal{L}}(\mathbf{B}, \mathbf{A}) = \mathrm{const} - \frac{1}{2(1 + \lambda)}\|\Pi_{\mathcal{H}(\mathbf{B}, \mathbf{A})}S\|_{L^2}^2,$$

where $S(\mathcal{X}) := \mathbb{E}[\boldsymbol{y} \mid \mathcal{X}]$ and $\mathcal{H}(\mathbf{B}, \mathbf{A})$ is the feature space induced by $(\mathbf{B}, \mathbf{A})$.

For any closed subspace $\mathcal{H} \subset L^2$, orthogonal projection satisfies

$$\|\Pi_{\mathcal{H}}S\|_{L^2}^2 \leq \|S\|_{L^2}^2,$$

with equality if and only if $S \in \mathcal{H}$. Hence $\widetilde{\mathcal{L}}(\mathbf{B}, \mathbf{A})$ is minimized if and only if

$$\|\Pi_{\mathcal{H}(\mathbf{B}, \mathbf{A})}S\|_{L^2}^2 = \|S\|_{L^2}^2 \quad \Longleftrightarrow \quad S \in \mathcal{H}(\mathbf{B}, \mathbf{A}).$$

By construction of the model and the true parameters,

$$S(\mathcal{X}) = \sum_{k \in K^\star} z_k^\star(\mathcal{X})\,(\mathbf{W}_k^\star)^\top \in \mathcal{H}^\star,$$

where $\mathcal{H}^\star$ is the feature space induced by $(\mathbf{B}^\star, \mathbf{A}^\star)$. Thus $S \in \mathcal{H}^\star$ and $\widetilde{\mathcal{L}}(\mathbf{B}^\star, \mathbf{A}^\star)$ attains the global minimum. Conversely, any global minimizer $(\bar{\mathbf{B}}, \bar{\mathbf{A}})$ must satisfy $S \in \mathcal{H}(\bar{\mathbf{B}}, \bar{\mathbf{A}})$.

Assumption 5 then implies that any two parameter tuples representing the same regression function $S$ are equivalent up to permutation and per-mode orthogonal rotations. Therefore the ground-truth parameters are identified within this natural equivalence class whenever the regression function $S$ is recovered.

**(II) Local Riemannian geometry.** We analyze the second-order behavior of $\widetilde{\mathcal{L}}(\mathbf{B}, \mathbf{A})$ near the true equivalence class of $(\mathbf{B}^\star, \mathbf{A}^\star)$. Since $\widetilde{\mathcal{L}}$ differs from $-\|\boldsymbol{\Sigma}_{z\boldsymbol{y}}\|_F^2$ only by an affine transform, it suffices to study $-\|\boldsymbol{\Sigma}_{z\boldsymbol{y}}\|_F^2$.

Consider the product manifold $\mathcal{M}$ consisting of all feasible $(\mathbf{B}, \mathbf{A})$ satisfying the global orthogonality constraints in Assumption 2. This is a product of generalized Stiefel manifolds and is compact. The equivalence transformations (permutations and per-mode orthogonal rotations) define an isometric group action on $\mathcal{M}$, and the true parameters form an orbit under this action. We work on the quotient manifold obtained by identifying points along this orbit (Absil et al., 2008).

At $(\mathbf{B}^\star, \mathbf{A}^\star)$, the gradient of $\widetilde{\mathcal{L}}$ vanishes in directions tangent to the quotient manifold, and we consider the Hessian restricted to the normal tangent space, which discards directions corresponding to the equivalence transformations. Let $\xi$ be a unit tangent vector in this normal space and let $t \mapsto (\mathbf{B}(t), \mathbf{A}(t))$ denote the geodesic with $(\mathbf{B}(0), \mathbf{A}(0)) = (\mathbf{B}^\star, \mathbf{A}^\star)$ and initial velocity $\dot{\mathbf{B}}(0), \dot{\mathbf{A}}(0)$ corresponding to $\xi$.

The per-mode projection matrices $\mathbf{M}_{i,p_i}(t)$ evolve along geodesics on the corresponding Stiefel manifolds. Standard perturbation results for orthogonal projectors (see Li (2006)) imply that the principal angles between $\mathrm{range}(\mathbf{M}_{i,p_i}(t))$ and $\mathrm{range}(\mathbf{M}_{i,p_i}^\star)$ grow linearly in $t$ for small $t$, and the alignment operators

$$\mathbf{T}_i(t) := \mathbf{M}_{i,p_i}(t)^\top \mathbf{M}_{i,p_i}^\star$$

satisfy

$$\|\mathbf{T}_i(t)\|_2 = 1 - c_i t^2 + o(t^2)$$

for some constants $c_i > 0$ depending on the direction $\xi$ and mode $i$, with $c_i = 0$ if $\xi$ lies entirely in the tangent space corresponding to rotations within the subspace, and $c_i > 0$ otherwise. Since we are on the normal space, at least one $c_i$ is strictly positive.

The overall alignment operator between the feature spaces is given by a Kronecker product $\bigotimes_{i=1}^n \mathbf{T}_i(t)$, and thus

$$\left\| \bigotimes_{i=1}^n \mathbf{T}_i(t) \right\|_2 = \prod_{i=1}^n \|\mathbf{T}_i(t)\|_2 = 1 - ct^2 + o(t^2)$$

for some $c > 0$ depending on the direction $\xi$. This quadratic decay of the alignment implies a quadratic decay of the projection energy $\|\Pi_{\mathcal{H}(\mathbf{B}(t), \mathbf{A}(t))} S\|_{L^2}^2$, and by Lemma 5 an associated quadratic increase in $\widetilde{\mathcal{L}}(\mathbf{B}(t), \mathbf{A}(t))$. Using Assumption 4, which ensures that the true regression function has nontrivial energy along each true component with strength at least $\alpha$, we obtain a uniform lower bound on the second derivative of $\widetilde{\mathcal{L}}$ along any unit normal direction:

$$\frac{d^2}{dt^2} \widetilde{\mathcal{L}}(\mathbf{B}(t), \mathbf{A}(t))\Big|_{t=0} \geq \frac{1}{1+\lambda} \min_{i \in [n]} \left( \alpha^2 \prod_{j \neq i} r_j \right) := \mu.$$

This establishes Riemannian $\mu$-strong convexity on the normal tangent space. Smoothness follows from the smooth dependence of $z$ and $\boldsymbol{\Sigma}_{zy}$ on $(\mathbf{B}, \mathbf{A})$ and the compactness of $\mathcal{M}$, which yields a finite Lipschitz constant $L$ for the Riemannian gradient of $\widetilde{\mathcal{L}}$.

**(III) Linear convergence of joint Riemannian GD.** The alternating algorithm consists of two steps:

(a) Given $(\mathbf{B}^{(t)}, \mathbf{A}^{(t)})$, update $\mathbf{W}$ to $\mathbf{W}^{(t+1)} := \mathbf{W}^\dagger(\mathbf{B}^{(t)}, \mathbf{A}^{(t)})$, which exactly minimizes $\mathcal{L}(\mathbf{B}^{(t)}, \mathbf{A}^{(t)}, \mathbf{W})$ with respect to $\mathbf{W}$ and hence does not increase the objective.

(b) With $\mathbf{W}$ fixed at $\mathbf{W}^{(t+1)}$, perform one Riemannian gradient descent step for $(\mathbf{B}, \mathbf{A})$ on the manifold $\mathcal{M}$ with respect to the reduced risk $\widetilde{\mathcal{L}}(\mathbf{B}, \mathbf{A})$, using a step size $\eta \in (0, 2/L)$.

By Part (II), in a neighborhood of the true equivalence class the reduced risk $\widetilde{\mathcal{L}}$ is Riemannian $\mu$-strongly convex and $L$-smooth on the normal tangent space of the quotient manifold. Standard results in Riemannian optimization (Absil et al., 2008) imply that Riemannian gradient descent with step size $\eta \in (0, 2/L)$ satisfies

$$\mathrm{dist}\big((\mathbf{B}^{(t+1)}, \mathbf{A}^{(t+1)}), (\mathbf{B}^\star, \mathbf{A}^\star)\big) \leq (1 - \eta\mu) \, \mathrm{dist}\big((\mathbf{B}^{(t)}, \mathbf{A}^{(t)}), (\mathbf{B}^\star, \mathbf{A}^\star)\big)$$

whenever $(\mathbf{B}^{(t)}, \mathbf{A}^{(t)})$ remains in the basin of strong convexity. Since the $\mathbf{W}$-update cannot increase $\mathcal{L}$ and does not move $(\mathbf{B}, \mathbf{A})$, the combined alternating scheme inherits this linear contraction in the distance to the true equivalence class, yielding the claimed convergence rate. □

### B.6 REASONABLENESS OF THE ASSUMPTIONS

Assumption 1 posits that the covariates follow a tensor-variate normal distribution with Kronecker-separable covariance, and that the noise is independent of $X$ with mean zero. Both parts are standard in the tensor-regression literature. Tensor normal models with separable Kronecker covariance are widely used in tensor-normal based regression (Llosa-Vite & Maitra, 2022; Guhaniyogi et al., 2017; Min et al., 2022). These works explicitly assume tensor normality with separable covariance for both covariates and errors. Independence between $X$ and the noise term, with

Gaussian errors, is the classical assumption in the normal linear model and its multivariate generalizations; it underlies essentially all standard theory for linear regression and least squares.

Assumptions 2–3 state that there exists a true IMOBT-type model: mode-wise projection matrices are orthonormal. In tensor regression, it is standard to assume that the true coefficient tensor has a specific low-rank structure (CP, Tucker, TT). For example, tensor regression works such as (Sun & Li, 2017; Qin & Zhu, 2025) impose low-rank and regularized conditions on the true factors to obtain uniqueness and statistical rates.

Assumption 4 requires that each active block has regression weights bounded away from zero. Intuitively, this means that the extracted components carry non-negligible predictive signals so that they can be distinguished from pure noise (Bühlmann & Van De Geer, 2011). These works require that nonzero coefficients are larger than a certain threshold to ensure that they are recoverable at the given sample size.

Assumption 5 states that, apart from permutations of components and orthogonal rotations within each latent subspace, the population IMOBT representation is unique. For classical block term decompositions (BTD), there are identifiability results under suitable Kruskal-type conditions (De Lathauwer & Nion, 2008). Our IMOBT model can be seen as an orthogonally constrained variant of such BTD models; therefore identifiability is preserved.

## C  GEOMETRY, PROJECTION/RETRACTION CORRECTNESS, CONVERGENCE, AND STATISTICAL GUARANTEES FOR RIEMANNIAN GRADIENT DESCENT IN OSCAR

This appendix provides explicit and self-contained derivations for the Riemannian optimization part of OSCAR. We use the same notation as in the main text. Moreover, this part of the content takes $\mathbf{B}$ as an example, and the optimization of the synergy matrix $\mathbf{A}$ is similar. To keep the formulas uncluttered, throughout this appendix we fix an input mode $i$ and a stage index $p_i$ and then drop these indices:

- $\mathbf{B} \equiv \mathbf{B}_{i,p_i} \in \mathbb{R}^{d \times b}$ denotes the spectral basis at mode $i$, where $d := d_i$ and $b := b_{i,p_i}$.
- $\mathbf{Q} \equiv \mathbf{Q}_{<p_i} \in \mathbb{R}^{d \times b_{\text{prev}}}$ denotes the concatenation of all previously extracted bases at mode $i$, where $b_{\text{prev}} := \sum_{q<p_i} b_{i,q}$. We assume that $\mathbf{Q}$ has orthonormal columns, i.e., $\mathbf{Q}^\top \mathbf{Q} = \mathbf{I}_{b_{\text{prev}}}$; in practice this is enforced once by a thin QR factorization.

The OSCAR constraints for the current block are

$$\mathbf{B}^\top \mathbf{B} = \mathbf{I}_b, \qquad \mathbf{B}^\top \mathbf{Q} = 0.$$

All norms are Frobenius unless otherwise stated.

### C.1  FEASIBLE SET AND TANGENT SPACE

For the fixed mode $i$ and stage $p_i$, the feasible set is the embedded submanifold

$$\mathcal{M} := \{ \mathbf{B} \in \mathbb{R}^{d \times b} : \mathbf{B}^\top \mathbf{B} = \mathbf{I}_b, \ \mathbf{B}^\top \mathbf{Q} = 0 \}.$$

The tangent space at a point $\mathbf{B} \in \mathcal{M}$ follows from linearizing the constraints. For a perturbation $\Delta \in \mathbb{R}^{d \times b}$, we have

$$(\mathbf{B} + \varepsilon\Delta)^\top (\mathbf{B} + \varepsilon\Delta) = \mathbf{I}_b + \varepsilon\,(\mathbf{B}^\top \Delta + \Delta^\top \mathbf{B}) + O(\varepsilon^2),$$

so differentiating $\mathbf{B}^\top \mathbf{B} = \mathbf{I}_b$ yields

$$\mathbf{B}^\top \Delta + \Delta^\top \mathbf{B} = 0.$$

Similarly,

$$(\mathbf{B} + \varepsilon\Delta)^\top \mathbf{Q} = \mathbf{B}^\top \mathbf{Q} + \varepsilon\,\Delta^\top \mathbf{Q} + O(\varepsilon^2)$$

and differentiating $\mathbf{B}^\top \mathbf{Q} = 0$ (with $\mathbf{Q}$ fixed at stage $p_i$) yields

$$\Delta^\top \mathbf{Q} = 0 \quad \Longleftrightarrow \quad \mathbf{Q}^\top \Delta = 0.$$

Hence

$$T_{\mathbf{B}}\mathcal{M} = \{ \Delta \in \mathbb{R}^{d \times b} : \mathbf{B}^\top \Delta + \Delta^\top \mathbf{B} = 0, \ \mathbf{Q}^\top \Delta = 0 \}.$$

### C.2  EXPLICIT ORTHOGONAL PROJECTION ONTO THE TANGENT SPACE

**Problem.**  Given a Euclidean gradient $\mathbf{G} \in \mathbb{R}^{d \times b}$, we seek its Euclidean orthogonal projection $\Pi_T(\mathbf{G})$ onto $T_{\mathbf{B}}\mathcal{M}$.

**Constrained least-squares formulation.** We solve

$$\min_{\mathbf{Z} \in \mathbb{R}^{d \times b}} \frac{1}{2} \|\mathbf{Z} - \mathbf{G}\|_F^2 \quad \text{s.t.} \quad \mathbf{B}^\top \mathbf{Z} + \mathbf{Z}^\top \mathbf{B} = 0, \ \mathbf{Q}^\top \mathbf{Z} = 0.$$

The Lagrangian is

$$\mathcal{L}(\mathbf{Z}, \mathbf{S}, \mathbf{M}) = \frac{1}{2} \|\mathbf{Z} - \mathbf{G}\|_F^2 + \langle \mathbf{S}, \ \mathbf{B}^\top \mathbf{Z} + \mathbf{Z}^\top \mathbf{B} \rangle + \langle \mathbf{M}, \ \mathbf{Q}^\top \mathbf{Z} \rangle,$$

where $\mathbf{S} \in \mathbb{R}^{b \times b}$ and $\mathbf{M} \in \mathbb{R}^{b_{\mathrm{prev}} \times b}$ are multipliers and $\langle A, B \rangle = \mathrm{tr}(A^\top B)$.

**Stationarity with respect to Z.** Taking the derivative with respect to $\mathbf{Z}$ and setting it to zero gives

$$0 = \nabla_{\mathbf{Z}} \mathcal{L} = (\mathbf{Z} - \mathbf{G}) + \mathbf{BS} + \mathbf{BS}^\top + \mathbf{QM}^\top.$$

Let $\mathbf{H} := \mathbf{S} + \mathbf{S}^\top$ (which is symmetric). Then

$$\mathbf{Z} = \mathbf{G} - \mathbf{BH} - \mathbf{QM}^\top.$$

**Imposing $\mathbf{Q}^\top \mathbf{Z} = 0$.** Left-multiplying by $\mathbf{Q}^\top$ yields

$$\mathbf{Q}^\top \mathbf{Z} = \mathbf{Q}^\top \mathbf{G} - \mathbf{Q}^\top \mathbf{BH} - \mathbf{Q}^\top \mathbf{QM}^\top.$$

Because $\mathbf{B}^\top \mathbf{Q} = 0$ implies $\mathbf{Q}^\top \mathbf{B} = 0$ and $\mathbf{Q}^\top \mathbf{Q} = \mathbf{I}_{b_{\mathrm{prev}}}$, we obtain

$$\mathbf{M}^\top = \mathbf{Q}^\top \mathbf{G} \quad \Longleftrightarrow \quad \mathbf{M} = \mathbf{G}^\top \mathbf{Q}.$$

**Imposing $\mathbf{B}^\top \mathbf{Z} + \mathbf{Z}^\top \mathbf{B} = 0$.** Using $\mathbf{Z} = \mathbf{G} - \mathbf{BH} - \mathbf{QM}^\top$ and the identities $\mathbf{B}^\top \mathbf{B} = \mathbf{I}_b$, $\mathbf{B}^\top \mathbf{Q} = 0$, $\mathbf{Q}^\top \mathbf{B} = 0$, we compute

$$\mathbf{B}^\top \mathbf{Z} + \mathbf{Z}^\top \mathbf{B} = \mathbf{B}^\top (\mathbf{G} - \mathbf{BH} - \mathbf{QM}^\top) + (\mathbf{G}^\top - \mathbf{H}^\top \mathbf{B}^\top - \mathbf{MQ}^\top)\mathbf{B}$$
$$= \mathbf{B}^\top \mathbf{G} - \mathbf{H} + \mathbf{G}^\top \mathbf{B} - \mathbf{H}^\top.$$

We require this to be zero, hence

$$\mathrm{sym}(\mathbf{B}^\top \mathbf{G}) - \mathrm{sym}(\mathbf{H}) = 0 \quad \Longrightarrow \quad \mathbf{H} = \mathrm{sym}(\mathbf{B}^\top \mathbf{G}),$$

where $\mathrm{sym}(\cdot) := \frac{1}{2}(\cdot + \cdot^\top)$.

**Final projection formula.** Substituting $\mathbf{H}$ and $\mathbf{M}$ into $\mathbf{Z}$ gives

$$\mathbf{Z}^\star = \mathbf{G} - \mathbf{B}\,\mathrm{sym}(\mathbf{B}^\top \mathbf{G}) - \mathbf{Q}\,\mathbf{Q}^\top \mathbf{G}.$$

Define the orthogonal projector onto the orthogonal complement of $\mathrm{span}(\mathbf{Q})$ by

$$P_\perp(\mathbf{Z}) := \mathbf{Z} - \mathbf{Q}(\mathbf{Q}^\top \mathbf{Z}) = (\mathbf{I} - \mathbf{QQ}^\top)\mathbf{Z}.$$

Since $\mathbf{B}^\top \mathbf{Q} = 0$, we have $\mathbf{B}^\top P_\perp(\mathbf{G}) = \mathbf{B}^\top \mathbf{G}$. Thus the projection can also be written as

$$\Pi_T(\mathbf{G}) = P_\perp(\mathbf{G}) - \mathbf{B}\,\mathrm{sym}(\mathbf{B}^\top P_\perp(\mathbf{G})).$$

Because the objective is strictly convex quadratic and the constraints are linear, this $\mathbf{Z}^\star$ is the unique Euclidean orthogonal projection onto $T_{\mathbf{B}}\mathcal{M}$. The Riemannian gradient is therefore

$$\mathrm{grad}_{\mathcal{M}} L(\mathbf{B}) = \Pi_T(\nabla L(\mathbf{B})),$$

where $\nabla L(\mathbf{B})$ denotes the Euclidean gradient of $L$ at $\mathbf{B}$.

### C.3 CORRECTNESS OF QR RETRACTION AND FIRST-ORDER AGREEMENT WITH THE GEODESIC

We adopt the standard QR retraction on the Stiefel manifold, restricted to the subspace orthogonal to $\mathbf{Q}$:

$$R(\mathbf{B}, \Xi) := \mathrm{qf}(\mathbf{B} + \Xi),$$

where qf returns the $\mathbf{Q}$ factor of the thin-QR decomposition with a positive diagonal in $\mathbf{R}$ (column-sign correction). Note that if $\mathbf{Q}^\top \Xi = 0$ and $\mathbf{B}^\top \mathbf{Q} = 0$, then $(\mathbf{B} + \Xi)^\top \mathbf{Q} = 0$ for all step sizes, so the column space of $\mathbf{B} + \Xi$ lies in the orthogonal complement of $\mathrm{span}(\mathbf{Q})$. Consequently, the orthonormal basis $\mathrm{qf}(\mathbf{B} + \Xi)$ also satisfies $R(\mathbf{B}, \Xi)^\top \mathbf{Q} = 0$, so $R(\mathbf{B}, \Xi) \in \mathcal{M}$.

**Retraction axiom 1.** Since $\mathbf{B} \in \mathcal{M}$ already has orthonormal columns and is orthogonal to $\mathbf{Q}$, its thin-QR factor is itself, so

$$R(\mathbf{B}, 0) = \mathrm{qf}(\mathbf{B}) = \mathbf{B}.$$

**Retraction axiom 2 (first-order agreement with the exponential map).** Let $\Xi \in T_\mathbf{B}\mathcal{M}$, i.e., $\mathbf{B}^\top \Xi + \Xi^\top \mathbf{B} = 0$ and $\mathbf{Q}^\top \Xi = 0$. Consider the curve $\mathbf{Y}(t) = \mathbf{B} + t\Xi$ and take a thin-QR factorization $\mathbf{Y}(t) = \mathbf{Q}(t)\mathbf{R}(t)$ with $\mathbf{Q}(t)^\top \mathbf{Q}(t) = \mathbf{I}_b$ and $\mathrm{diag}(\mathbf{R}(t)) > 0$. A standard first-order expansion (see, e.g., Absil, Mahony, and Sepulchre, *Optimization Algorithms on Matrix Manifolds*) yields

$$\mathbf{R}(t) = \mathbf{I}_b + \mathrm{sym}(\mathbf{B}^\top t\Xi) + o(t) = \mathbf{I}_b + o(t),$$

because $\mathbf{B}^\top \Xi$ is skew-symmetric by tangency. Hence

$$\mathbf{Q}(t) = \mathbf{Y}(t)\mathbf{R}(t)^{-1} = (\mathbf{B} + t\Xi)(\mathbf{I}_b + o(t)) = \mathbf{B} + t\Xi + o(t).$$

Therefore

$$\left.\frac{d}{dt}\right|_{t=0} R(\mathbf{B}, t\Xi) = \left.\frac{d}{dt}\right|_{t=0} \mathbf{Q}(t) = \Xi,$$

showing that $R$ is a valid retraction on $\mathcal{M}$ and is first-order equivalent to the Riemannian exponential map along $\Xi$.

### C.4 RIEMANNIAN PL GEOMETRY AND LINEAR CONVERGENCE OF RGD

We briefly recall standard relations between Riemannian smoothness, the Polyak–Łojasiewicz (PL) condition, and the linear convergence of Riemannian gradient descent (RGD) (Boumal, 2023; Karimi et al., 2016).

Let $f : \mathcal{M} \to \mathbb{R}$ be the empirical risk on the product manifold $\mathcal{M}$, equipped with the canonical Riemannian metric. We say that $f$ is (locally) Riemannian $L$-smooth if, for any $x \in \mathcal{M}$, any $v \in T_x\mathcal{M}$, and any retraction curve $\gamma(t) = R(x, tv)$,

$$f(\gamma(t)) \leq f(x) + t\langle \mathrm{grad}f(x), v\rangle + \tfrac{L}{2} t^2 \|v\|^2. \tag{12}$$

Here $\mathrm{grad}f(x)$ denotes the Riemannian gradient at $x$.

Taking one Riemannian gradient step $x_+ = R(x, -\eta g)$ with $g = \mathrm{grad}f(x)$ and stepsize $\eta > 0$, and applying equation 12 with $v = -\eta g$, yields the standard descent inequality

$$f(x_+) \leq f(x) - \left(\eta - \tfrac{L}{2}\eta^2\right)\|g\|^2. \tag{13}$$

In particular, $f(x_+) \leq f(x)$ whenever $\eta \in (0, 2/L]$.

A local Riemannian PL condition with constant $\mu > 0$ on a neighborhood $\mathcal{U}$ of the solution set is the inequality

$$\tfrac{1}{2}\|\mathrm{grad}f(x)\|^2 \geq \mu\left(f(x) - f^\star\right), \qquad \forall x \in \mathcal{U}. \tag{14}$$

Combining equation 13 and equation 14 gives, for any $x \in \mathcal{U}$,

$$f(x_+) - f^\star \leq \rho(\eta)\left(f(x) - f^\star\right), \qquad \rho(\eta) := 1 - 2\mu\left(\eta - \tfrac{L}{2}\eta^2\right).$$

For any $\eta \in (0, 2/L)$ we have $\rho(\eta) \in (0, 1)$, and choosing $\eta = 1/L$ yields the contraction factor $\rho = 1 - \mu/L$. Thus, under local $L$-smoothness and a local PL condition, fixed-stepsize RGD converges linearly as long as the iterates remain in $\mathcal{U}$.

Conversely, up to constants, a PL-type inequality is essentially necessary for local linear convergence of fixed-stepsize RGD. More precisely, suppose that for some fixed $\eta \in (0, 1/L]$ and $\rho \in (0, 1)$ the RGD iterates satisfy

$$f(x_{t+1}) - f^\star \leq \rho\left(f(x_t) - f^\star\right)$$

for all iterates in a neighborhood of the solution set. Combining this assumption with equation 13 yields

$$f(x_t) - f(x_{t+1}) \geq \left(\eta - \tfrac{L}{2}\eta^2\right)\|\mathrm{grad}f(x_t)\|^2 \geq (1 - \rho)\left(f(x_t) - f^\star\right),$$

and hence

$$\tfrac{1}{2}\|\mathrm{grad}f(x_t)\|^2 \geq \mu_{\mathrm{PL}}\left(f(x_t) - f^\star\right), \qquad \mu_{\mathrm{PL}} := \frac{1 - \rho}{2(\eta - \tfrac{L}{2}\eta^2)} > 0.$$

See (Karimi et al., 2016) for the Euclidean case and Boumal (2023) for Riemannian extensions. We use this equivalence only as a convenient geometric characterization of the local optimization landscape; we do *not* claim that our OSCAR loss satisfies a global PL condition.

### C.5 PRECONDITIONED STEPS AS RGD IN A MODIFIED METRIC

We next recall a standard descent estimate for preconditioned Riemannian gradient steps, which can be viewed as RGD under a modified metric (Boumal, 2023).

Let $\mathbf{P} \succ 0$ be a symmetric positive definite preconditioner acting in the tangent space $T_x\mathcal{M}$. Given the Riemannian gradient $g = \mathrm{grad} f(x)$, consider the update

$$v \ = \ -\eta\,\mathbf{P}^{-1}g, \qquad x_+ \ = \ R(x, v).$$

Applying $L$-smoothness equation 12 along $\gamma(t) = R(x, tv)$ gives

$$f(x_+) \leq f(x) + \langle g, v \rangle + \tfrac{L}{2}\|v\|^2$$
$$= f(x) - \eta\,\langle g, \mathbf{P}^{-1}g \rangle + \tfrac{L}{2}\eta^2\|\mathbf{P}^{-1}g\|^2.$$

Assume spectral bounds $0 < m \leq \lambda_{\min}(\mathbf{P}) \leq \lambda_{\max}(\mathbf{P}) \leq M$. Then

$$\langle g, \mathbf{P}^{-1}g \rangle \ \geq \ \frac{1}{M}\|g\|^2, \qquad \|\mathbf{P}^{-1}g\| \ \leq \ \frac{1}{m}\|g\|.$$

Therefore

$$f(x_+) - f(x) \ \leq \ -\Big(\frac{\eta}{M} - \frac{L}{2}\frac{\eta^2}{m^2}\Big)\|g\|^2. \tag{15}$$

In particular, $f$ strictly decreases whenever $0 < \eta \leq m^2/(LM)$.

If, in addition, a local PL inequality equation 14 holds with constant $\mu > 0$, then combining equation 14 and equation 15 yields

$$f(x_+) - f^\star \ \leq \ \Big[1 - 2\mu\frac{\eta}{M} + \mu L\frac{\eta^2}{m^2}\Big]\big(f(x) - f^\star\big).$$

Minimizing the quadratic bracket over $\eta$ gives the optimal constant stepsize $\eta^\star = m^2/(LM)$ and the corresponding linear rate

$$\rho^\star \ = \ 1 - \frac{\mu\,m^2}{L\,M^2}.$$

Thus, under a local PL condition and bounded condition number $M/m$, preconditioning preserves linear convergence and affects only the rate constants, not the qualitative behavior.

### C.6 STAGEWISE LEARNING WITH COLLABORATIVE REFINEMENT: MONOTONICITY

OSCAR learns components sequentially while enforcing orthogonality in parameter space and allowing collaborative refinement of synergy matrices and the regression head at each stage.

Let $\mathcal{F}^{(p)}$ denote the feasible set at stage $p$ (i.e., after extracting $p$ components) and define

$$F_p := \min_{\Theta \in \mathcal{F}^{(p)}} \mathcal{L}(\Theta),$$

where $\mathcal{L}$ is the (empirical or population) loss. The feasible set at stage $p$ contains a copy of the stage-$(p-1)$ solution: we can embed any $\Theta^{(p-1)} \in \mathcal{F}^{(p-1)}$ into $\mathcal{F}^{(p)}$ by setting the new block's contribution to zero and keeping all previously learned parameters unchanged. Therefore

$$\min_{\Theta \in \mathcal{F}^{(p)}} \mathcal{L}(\Theta) \ \leq \ \mathcal{L}\big(\text{embedded old solution}\big) \ = \ F_{p-1},$$

and, after re-optimizing the synergy matrices and the head at stage $p$, we obtain

$$F_p \ \leq \ F_{p-1}.$$

Thus the optimal value is monotonically nonincreasing across stages. We do not claim any formal result on the evolution of local PL geometry across stages; the stagewise design in OSCAR is primarily motivated by empirical performance and interpretability.

### C.7 Summary of Geometric Guarantees

For convenience, we summarize the geometric and algorithmic properties:

(i) **Tangent-space projection with cross-component orthogonality.** For a fixed mode and stage, the feasible set for $\mathbf{B}$ is an embedded submanifold defined by column-orthogonality and orthogonality to previously learned bases. We derived the explicit Euclidean orthogonal projection of an arbitrary matrix $\mathbf{G}$ onto the tangent space:

$$\Pi_T(\mathbf{G}) = \underbrace{\mathbf{G} - \mathbf{Q}(\mathbf{Q}^\top \mathbf{G})}_{P_\perp(\mathbf{G})} - \mathbf{B}\operatorname{sym}(\mathbf{B}^\top P_\perp(\mathbf{G})),$$

where $\mathbf{Q}$ collects previous bases and $\operatorname{sym}(\cdot) := \frac{1}{2}(\cdot + \cdot^\top)$..

(ii) **QR retraction.** The mapping $R(\mathbf{B}, \Xi) = \operatorname{qf}(\mathbf{B} + \Xi)$, the $\mathbf{Q}$ factor of the thin QR decomposition, is a valid retraction on the constraint manifold and is first-order equivalent to the Riemannian exponential map along any tangent direction $\Xi$.

(iii) **RGD descent and PL-based linear convergence (background).** Under local Riemannian $L$-smoothness, Riemannian gradient descent with stepsize $\eta \in (0, 2/L]$ satisfies the descent inequality equation 13. If, in addition, the loss satisfies a local Riemannian PL condition with constant $\mu > 0$, then fixed-stepsize RGD converges linearly with contraction factor $1 - 2\mu(\eta - \frac{L}{2}\eta^2)$ (in particular $1 - \mu/L$ for $\eta = 1/L$). Conversely, local linear convergence of fixed-stepsize RGD implies a PL-type inequality with an explicit constant, as in Karimi et al. (2016); Boumal (2023). These are standard results that we recall only for completeness.

(iv) **Preconditioned RGD (background).** For preconditioned updates with a symmetric positive definite $\mathbf{P}$ satisfying $m\mathbf{I} \preceq \mathbf{P} \preceq M\mathbf{I}$, the loss decreases for any stepsize $0 < \eta \leq m^2/(LM)$ and, under a local PL condition, the iterates converge linearly with rate $1 - (\mu m^2)/(LM^2)$. This shows that well-conditioned preconditioners preserve the qualitative convergence behavior.

(v) **Stagewise monotonicity.** In the stagewise OSCAR procedure, the feasible set enlarges with each new component. As a result, the optimal value $F_p$ at stage $p$ is monotonically nonincreasing in $p$. OSCAR does not rely on any stronger geometric guarantees across stages.

# D Detailed Derivation of the SCRGD Update

This section provides the detailed steps for the Subspace Constrained Riemannian Gradient Descent (SCRGD) update used to optimize the spectral basis matrices $\mathbf{B}_{i,p}$ in OSCAR. The presentation is consistent with the Riemannian geometry in Appendix C and with the pseudocode in the main text.

## D.1 Constraint Manifold and Notation

For a fixed input mode $i$ and stage $p$, we optimize the basis $\mathbf{B}_{i,p} \in \mathbb{R}^{d_i \times b_{i,p}}$ subject to

$$\mathbf{B}_{i,p}^\top \mathbf{B}_{i,p} = \mathbf{I}_{b_{i,p}}, \qquad \mathbf{B}_{i,p}^\top \mathbf{Q}_{<p} = 0,$$

where

$$\mathbf{Q}_{<p} := \begin{bmatrix} \mathbf{B}_{i,1}, \ldots, \mathbf{B}_{i,p-1} \end{bmatrix} \in \mathbb{R}^{d_i \times b_{\mathrm{prev}}}$$

is the concatenation of all previously learned bases on mode $i$, and we assume $\mathbf{Q}_{<p}$ has orthonormal columns ($\mathbf{Q}_{<p}^\top \mathbf{Q}_{<p} = \mathbf{I}$) after an initial QR orthogonalization.

The feasible set for $\mathbf{B}_{i,p_i}$ is the embedded submanifold

$$\mathcal{M}_{i,p} := \left\{ \mathbf{B} \in \mathbb{R}^{d_i \times b_{i,p}} \ : \ \mathbf{B}^\top \mathbf{B} = \mathbf{I}, \ \mathbf{B}^\top \mathbf{Q}_{<p} = 0 \right\}.$$

As in Appendix C, the tangent space at $\mathbf{B} \in \mathcal{M}_{i,p}$ is

$$T_\mathbf{B} \mathcal{M}_{i,p} = \left\{ \Delta \in \mathbb{R}^{d_i \times b_{i,p}} : \mathbf{B}^\top \Delta + \Delta^\top \mathbf{B} = 0, \ \mathbf{Q}_{<p}^\top \Delta = 0 \right\}.$$

For conciseness, throughout this section we fix $(i, p)$ and write $\mathbf{B} \equiv \mathbf{B}_{i,p}$ and $\mathbf{Q} \equiv \mathbf{Q}_{<p}$, with $d := d_i$ and $b := b_{i,p}$.

## D.2 ONE SCRGD ITERATION

A single SCRGD iteration for $B$ consists of three steps, applied to minimize the overall loss

$$\mathcal{L}(\Theta) = \frac{1}{2N} \sum_{j=1}^{N} \left\| \mathcal{Y}_j - \widehat{\mathcal{Y}}_j \right\|_F^2.$$

**Step 1: Euclidean gradient.** Using standard backpropagation, we compute the Euclidean gradient of the loss with respect to $\mathbf{B}$,

$$\mathbf{G} := \nabla_{\mathbf{B}} \mathcal{L}(\Theta) \in \mathbb{R}^{d \times b}.$$

This gradient lives in the ambient Euclidean space and does not in general respect the constraints defining $\mathcal{M}_{i,p}$.

**Step 2: Projection onto the tangent space.** To obtain a valid Riemannian search direction, we project $\mathbf{G}$ onto $T_{\mathbf{B}} \mathcal{M}_{i,p}$ using the explicit orthogonal projection derived in Appendix C. This is conveniently implemented in two sequential substeps:

(a) **Remove components in the span of $\mathbf{Q}$.** First project $\mathbf{G}$ onto the orthogonal complement of $\mathrm{span}(\mathbf{Q})$ to enforce the inter-component orthogonality constraint:

$$\mathbf{G}_\perp := \mathbf{G} - \mathbf{Q}(\mathbf{Q}^\top \mathbf{G}) = (\mathbf{I} - \mathbf{Q}\mathbf{Q}^\top)\mathbf{G}. \tag{16}$$

By construction $\mathbf{Q}^\top \mathbf{G}_\perp = 0$.

(b) **Project onto the Stiefel tangent.** Next, project $\mathbf{G}_\perp$ onto the tangent space of the Stiefel manifold at $\mathbf{B}$ to enforce the intra-component orthonormality constraint $\mathbf{B}^\top \mathbf{B} = \mathbf{I}$:

$$\mathrm{grad}_{\mathbf{B}} \mathcal{L} := \mathbf{G}_\perp - \mathbf{B}\,\mathrm{sym}(\mathbf{B}^\top \mathbf{G}_\perp), \qquad \mathrm{sym}(\mathbf{A}) := \tfrac{1}{2}(\mathbf{A} + \mathbf{A}^\top). \tag{17}$$

Combining equation 16 and equation 17, the full orthogonal projection of $\mathbf{G}$ onto $T_{\mathbf{B}} \mathcal{M}_{i,p}$ is

$$\Pi_{T_{\mathbf{B}} \mathcal{M}_{i,p}}(\mathbf{G}) = \mathbf{G} - \mathbf{Q}(\mathbf{Q}^\top \mathbf{G}) - \mathbf{B}\,\mathrm{sym}(\mathbf{B}^\top(\mathbf{G} - \mathbf{Q}(\mathbf{Q}^\top \mathbf{G}))),$$

and we take the Riemannian gradient as

$$\mathrm{grad}_{\mathcal{M}} \mathcal{L}(\mathbf{B}) := \Pi_{T_{\mathbf{B}} \mathcal{M}_{i,p}}(\mathbf{G}).$$

This formula is exactly the projection established in Appendix C, and it coincides with the implementation in the pseudocode (which first computes $\mathbf{G}_\perp$ and then applies the Stiefel projection).

**Step 3: Retraction.** We move along the negative Riemannian gradient and retract back to the manifold using a QR-based retraction. Given a stepsize $\eta > 0$, we form the tentative update

$$\widetilde{\mathbf{B}} := \mathbf{B} - \eta\,\mathrm{grad}_{\mathcal{M}} \mathcal{L}(\mathbf{B}),$$

and then compute a thin QR factorization

$$\widetilde{\mathbf{B}} = \mathbf{Q}_{\mathbf{B}} \mathbf{R}_{\mathbf{B}},$$

with $\mathbf{Q}_{\mathbf{B}}^\top \mathbf{Q}_{\mathbf{B}} = \mathbf{I}$ and $\mathrm{diag}(\mathbf{R}_{\mathbf{B}})$ chosen with positive entries (column-sign correction). We set

$$\mathbf{B}^{\mathrm{new}} := \mathbf{Q}_{\mathbf{B}}.$$

As shown in Appendix C, the mapping $\mathbf{R}(\mathbf{B}, \Xi) := \mathrm{qf}(\mathbf{B} + \Xi)$ defines a valid retraction on $\mathcal{M}_{i,p}$ and is first-order equivalent to the Riemannian exponential map along any tangent direction $\Xi$.

Putting these steps together, one SCRGD iteration for $\mathbf{B}_{i,p}$ is:

$$\mathbf{B}_{i,p}^{(t+1)} = \mathrm{qf}\left(\mathbf{B}_{i,p}^{(t)} - \eta\,\mathrm{grad}_{\mathcal{M}} \mathcal{L}(\mathbf{B}_{i,p}^{(t)})\right),$$

with $\mathrm{grad}_{\mathcal{M}} \mathcal{L}$ given by equation 16–equation 17. This is precisely the update implemented in the SCRGD pseudocode in the main text, and it rigorously maintains both intra-component and inter-component orthogonality throughout the optimization.

### D.3 INITIALIZATION

The feature extraction matrix $\mathbf{B}$ is initialized using an orthogonal basis constructed from the discrete Fourier transform (DFT), rather than with a random initialization. This choice allows the starting point to emphasize low-frequency components, which generally contain the most significant structural information in the data, and therefore provides a more informative and stable initialization for subsequent optimization.

## E   DATA PROCESSING DETAILS

### E.1   SYNTHETIC DATA GENERATION.

We first generate a dense coefficient tensor $\mathcal{B}^\star$ with i.i.d. standard Gaussian entries. Then we approximate $\mathcal{B}^\star$ by our IMOBT low-rank structure using an alternating least squares (ALS) procedure. Concretely, ALS iteratively updates the block-structured factor matrices and the core tensor to minimize the reconstruction loss, yielding a low-rank representation $\widetilde{\mathcal{B}}$ that has the same dimensions as $\mathcal{B}^\star$ but is constrained to the IMOBT model family. This $\widetilde{\mathcal{B}}$ is treated as the "true" coefficient tensor in all simulations.

The i.i.d. input tensors $\{\mathcal{X}_s\}_{s=1}^N$ are generated with entries drawn independently from a standard Gaussian distribution. For each $\mathcal{X}_s$, the observed output is obtained by contracting $\mathcal{X}_s$ with $\widetilde{\mathcal{B}}$ and adding a standard Gaussian noise:

$$\mathcal{Y}_s \;=\; \left\langle \mathcal{X}_s, \widetilde{\mathcal{B}} \right\rangle_{1:n} + \mathcal{E}_s, \quad \mathcal{E}_s \sim \mathcal{N}(0, \sigma^2 \mathbf{I}). \tag{18}$$

Repeating this procedure for $s = 1, \ldots, N$ yields the synthetic dataset $\left\{ (\mathcal{X}_s, \mathcal{Y}_s) \right\}_{s=1}^N$, which is exactly sampled from an IMOBT model with known low-rank ground-truth coefficients $\widetilde{\mathcal{B}}$.

### E.2   ECOG DATASET PREPROCESSING

This section provides the full, step-by-step preprocessing pipeline used to construct the samples for the ECoG Monkey Dataset from the raw recordings, as mentioned in the main experimental setup.

1. **Filtering and Referencing:** The raw 64-channel ECoG signals, sampled at 1000 Hz, are first band-pass filtered between 0.3 Hz and 500 Hz using a fourth-order Butterworth filter. Subsequently, a Common Average Reference (CAR) is applied to remove global noise.

2. **Spectrogram Generation:** We compute the power spectrogram for each channel using a continuous Morlet wavelet transform ($w = 6.0$). This is performed across 10 linearly spaced frequency bins ranging from 10 Hz to 120 Hz.

3. **Normalization:** The resulting power values are z-scored independently for each frequency bin across the time dimension to normalize their distributions.

4. **Tensor Construction:** For each motion capture timestamp (sampled at 120 Hz), we form a predictor tensor $\mathcal{X}_i \in \mathbb{R}^{64 \times 10 \times 10}$ (Channels × Frequencies × Lags). This tensor represents the normalized spectral power from 10 historical time lags, spanning 0.1 to 1.0 seconds prior to the prediction time.

5. **Response Vector:** The corresponding response vector $\mathbf{y}_i \in \mathbb{R}^3$ is the 3D coordinate of the monkey's right wrist, transformed into a body-centered reference frame defined by the shoulders.

6. **Artifact Removal and Splitting:** Samples contaminated by chewing artifacts are identified and excluded. The cleaned dataset is then split chronologically: the first 10 minutes of data are used for training, and the subsequent 5 minutes serve as the test set.

## F   PRELIMINARY OF LOW-RANK TENSOR DECOMPOSITIONS

To make the estimation of $\mathcal{B}$ feasible, a low-rank structure is typically imposed.

- **Canonical PARAFAC (CP) Decomposition:** This decomposition models a tensor as a sum of $R$ rank-one tensors:

$$\mathcal{B} \approx \sum_{r=1}^R \boldsymbol{b}_1^{(r)} \circ \cdots \circ \boldsymbol{b}_{n+m}^{(r)}, \tag{19}$$

where $\circ$ denotes the vector outer product. The smallest such $R$ is the rank of the tensor.

- **Tucker Decomposition:** This method decomposes a tensor into a (typically small) core tensor $\mathcal{G}$ and a set of factor matrices $\{\mathbf{B}^{(k)}\}$ along each mode:

$$\mathcal{B} \approx \mathcal{G} \times_1 \mathbf{B}^{(1)} \times_2 \cdots \times_{n+m} \mathbf{B}^{(n+m)}. \tag{20}$$

The factor matrices are often constrained to have orthonormal columns to ensure model identifiability.

- **Tensor Train (TT) Decomposition:** This decomposition represents a tensor as a sequence of 3-way cores $\{\mathcal{G}^{(k)}\}_{k=1}^{n+m}$ with TT-ranks $(r_0, r_1, \ldots, r_{n+m})$ (with $r_0 = r_{n+m} = 1$). Each core $\mathcal{G}^{(k)} \in \mathbb{R}^{r_{k-1} \times d_k \times r_k}$ has one physical mode of size $d_k$ and two rank modes. The entries of $\mathcal{B}$ are approximated as

$$\mathcal{B}(i_1, \ldots, i_{n+m}) \approx \sum_{\alpha_1, \ldots, \alpha_{n+m-1}} \mathcal{G}^{(1)}(1, i_1, \alpha_1) \mathcal{G}^{(2)}(\alpha_1, i_2, \alpha_2) \cdots \mathcal{G}^{(n+m)}(\alpha_{n+m-1}, i_{n+m}, 1), \tag{21}$$

where the TT-ranks $\{r_k\}$ control the degree of compression: small ranks yield a compact low-rank representation of $\mathcal{B}$.

# G  DEFINITION OF COMBINATION TUPLE

## NOTATION

Let $n \in \mathbb{N}$ be the number of modes. For each mode $i = 1, \ldots, n$, let the maximum number of PCA blocks be $K_i \in \mathbb{N}$.

The full index set of all PCA block combinations is

$$\mathcal{P}_{\text{full}} = \big\{ \mathbf{p} = (p_1, \ldots, p_n) \in \mathbb{Z}^n \mid 1 \le p_i \le K_i \text{ for all } i \big\}.$$

Define the endpoints of a line segment in $\mathbb{R}^n$ by $\mathbf{P}_1 = (1, \ldots, 1)^\top$ and $\mathbf{P}_2 = (K_1, \ldots, K_n)^\top$, with direction vector $\mathbf{v} = \mathbf{P}_2 - \mathbf{P}_1$. Let $\rho > 0$ be the distance threshold (the value of diag).

For any $\mathbf{p} \in \mathcal{P}_{\text{full}}$, set $t^*(\mathbf{p}) = \langle \mathbf{p} - \mathbf{P}_1, \mathbf{v} \rangle / \|\mathbf{v}\|_2^2$, $\tilde{t}(\mathbf{p}) = \min\{1, \max\{0, t^*(\mathbf{p})\}\}$, $\text{proj}(\mathbf{p}) = \mathbf{P}_1 + \tilde{t}(\mathbf{p})\,\mathbf{v}$, and distance $d(\mathbf{p}) = \|\mathbf{p} - \text{proj}(\mathbf{p})\|_2$.

The set of combinations selected by diag is

$$\mathcal{P} = \big\{ \mathbf{p} \in \mathcal{P}_{\text{full}} \mid d(\mathbf{p}) \le \rho \big\}.$$

## NON-SEQUENTIAL MODE

In OSCAR-IMOBT low rank model, all selected combinations in $\mathcal{C}$ are always active:

$$\mathcal{P}_t^{(\text{NS})} = \mathcal{P} \quad \text{for every training iteration } t.$$

## SEQUENTIAL MODE

Let $K_{\max} = \max_{1 \le i \le n} K_i$. For each stage $p \in \{1, \ldots, K_{\max}\}$, define the active set

$$\mathcal{P}^{(\text{S})}(p) = \big\{ \mathbf{p} = (p_1, \ldots, p_n) \in \mathcal{P} \mid \max_{1 \le j \le n} p_j \le p \big\}.$$

In sequential mode, at stage $p$ the model only uses the combinations in $\mathcal{P}^{(\text{S})}(p)$, and $p$ is increased step by step from 1 up to $K_{\max}$.

# H  EXPERIMENTAL SETTINGS

**Study 1 (Table 1) - OSCAR-IMOBT Low-Rank Model:** For the **ECOG** dataset, we set the block sizes to $\mathbf{b} = (5, 2, 2)$ and projection dimensions to $\mathbf{P} = (5, 2, 2)$, with diag of 0.5. For the **LFW** dataset, we set $\mathbf{b} = (15, 15, 1)$ and $\mathbf{P} = (4, 4, 3)$, with diag of 1.0.

---

**Algorithm 3** Riemannian Update for $A_{i,k}$ on Stiefel Manifold

---

**Require:** Current synergy matrix $\mathbf{A}_{i,k} \in \mathrm{St}(b_i, r_i)$; Euclidean gradient $\mathbf{G_A} \in \mathbb{R}^{b_i \times r_i}$; step size $\eta_A > 0$.

1: **Tangent projection (Riemannian gradient):**

$$\mathbf{G}_{A,\mathrm{riem}} = \mathbf{G}_A - \mathbf{A}_{i,k}\,\mathrm{sym}(\mathbf{A}_{i,k}^\top \mathbf{G_A}), \quad \mathrm{sym}(\mathbf{M}) = \tfrac{1}{2}(\mathbf{M} + \mathbf{M}^\top).$$

2: **Gradient step in the tangent space:**

$$\tilde{\mathbf{A}}_{i,k} = \mathbf{A}_{i,k} - \eta_{\mathbf{A}}\,\mathbf{G}_{\mathbf{A},\mathrm{riem}}.$$

3: **Retraction via QR:** compute the thin QR factorization $\tilde{\mathbf{A}}_{i,k} = \mathbf{QR}$ and set $\mathbf{A}_{i,k}^+ \leftarrow \mathbf{Q}$.

4: **return** Updated $\mathbf{A}_{i,k}^+ \in \mathrm{St}(b_i, r_i)$.

---

**Algorithm 4** Joint Riemannian Update for Mutually Orthogonal Bases $\{\mathbf{B}_{i,k}\}_{k=1}^K$

---

**Require:** Current bases $\{\mathbf{B}_{i,k} \in \mathbb{R}^{d_i \times b_i}\}_{k=1}^K$; Euclidean gradients $\{\mathbf{G}_{B,k} \in \mathbb{R}^{d_i \times b_i}\}_{k=1}^K$; step size $\eta_B > 0$.

1: **Stack bases and gradients:**

$$\mathbf{Q} = [\mathbf{B}_{i,1}, \ldots, \mathbf{B}_{i,K}] \in \mathbb{R}^{d_i \times Kb_i}, \quad \mathbf{G}_B = [\mathbf{G}_{B,1}, \ldots, \mathbf{G}_{B,K}] \in \mathbb{R}^{d_i \times Kb_i}.$$

2: **Riemannian gradient on** $\mathrm{St}(d_i, Kb_i)$**:**

$$\mathbf{G}_{B,\mathrm{riem}} = \mathbf{G}_B - \mathbf{Q}\,\mathrm{sym}(\mathbf{Q}^\top \mathbf{G}_B), \quad \mathrm{sym}(\mathbf{M}) = \tfrac{1}{2}(\mathbf{M} + \mathbf{M}^\top).$$

3: **Gradient step:**

$$\tilde{\mathbf{Q}} = \mathbf{Q} - \eta_B\,\mathbf{G}_{B,\mathrm{riem}}.$$

4: **Retraction via QR:** compute the thin QR factorization $\tilde{\mathbf{Q}} = \mathbf{UR}$ and set $\mathbf{Q}^+ \leftarrow \mathbf{U}$.

5: **Unstack updated bases:** partition $\mathbf{Q}^+ = [\mathbf{B}_{i,1}^+, \ldots, \mathbf{B}_{i,K}^+]$ with each $\mathbf{B}_{i,k}^+ \in \mathbb{R}^{d_i \times b_i}$.

6: **return** Updated bases $\{\mathbf{B}_{i,k}^+\}_{k=1}^K$ satisfying $\mathbf{Q}^{+\top}\mathbf{Q}^+ = I_{Kb_i}$, i.e., all columns in $\{\mathbf{B}_{i,k}^+\}$ are mutually orthonormal (up to numerical error).

---

**Study 2 (Table 2) and Figure 4 - OSCAR Model:** For experiments involving the specific introduction of $A_{i,k}$ (including Table 2 and Fig. 4), the number of components was set to $K = 4$. The structural hyperparameters were configured as follows: block sizes $\mathbf{b} = (8, 3, 3)$, ranks $\mathbf{r} = (5, 2, 2)$, and projection dimensions $\mathbf{P} = (4, 3, 3)$, with a diagonal parameter of $0.5$.

**Optimization and Regularization Details:** Consistent across all experiments, the regularization strength for $W$ was set to $10^{-4}$. The optimization was performed using back-propagation with a learning rate of $5 \times 10^{-4}$. The maximum number of epochs was set to 10,000, consistent with the settings used for the baseline models utilizing back-propagation. To prevent overfitting, we employed early stopping with a tolerance of $10^{-7}$ and a patience of 50 epochs.

## I    RIEMANNIAN UPDATE RULES FOR $A_{i,k}$ AND $\mathbf{B}_{i,k}$

Algorithms 35– summarize the Riemannian optimization routines used in our model. Algorithm 3 performs a standard Riemannian gradient step with QR-based retraction for updating the synergy matrix $A_{i,k}$ on the Stiefel manifold. Algorithm 4 jointly updates all bases $\{\mathbf{B}_{i,k}\}_{k=1}^K$ on a larger Stiefel manifold so that their columns remain mutually orthonormal. Algorithm 5 provides a subspace-constrained variant that updates each $\mathbf{B}_{i,k}$ sequentially in the orthogonal complement of the previously learned bases, enforcing cross-component orthogonality in a staged manner.

## J    GRID SEARCH OVER RANK CONFIGURATIONS

Figure 5 illustrates the grid search procedure used to select the tensor ranks. For each candidate rank configuration within a moderate range determined by the data dimensions, we train the model and evaluate the validation loss, yielding a performance landscape over the rank grid. The highlighted region indicates a broad operating regime in which the method attains a favorable trade-off between model capacity, reconstruction error, and computational cost.

---

**Algorithm 5** Subspace-Constrained Riemannian Update for $\mathbf{B}_{i,k}$ (SCRGD)

---

**Require:** Current basis $\mathbf{B}_{i,k} \in \mathbb{R}^{d_i \times b_i}$; previously learned bases $\mathbf{Q}_{<k} = [\mathbf{B}_{i,1}, \ldots, \mathbf{B}_{i,k-1}]$; Euclidean gradient $\mathbf{G}_\mathbf{B} \in \mathbb{R}^{d_i \times b_i}$; step size $\eta_B > 0$.

1: **Project gradient to orthogonal complement of** $\mathrm{span}(\mathbf{Q}_{<k})$**:**

$$\mathbf{G}_\perp = \left(\mathbf{I} - \mathbf{Q}_{<k}\mathbf{Q}_{<k}^\top\right)\mathbf{G}_B.$$

2: **Riemannian projection on Stiefel (tangent space at $\mathbf{B}_{i,k}$):**

$$\mathbf{G}_{B,\mathrm{riem}} = \mathbf{G}_\perp - \mathbf{B}_{i,k}\,\mathrm{sym}(\mathbf{B}_{i,k}^\top\mathbf{G}_\perp).$$

3: **Gradient step:**

$$\tilde{\mathbf{B}}_{i,k} = \mathbf{B}_{i,k} - \eta_\mathbf{B}\,\mathbf{G}_{\mathbf{B},\mathrm{riem}}.$$

4: **Retraction via QR:** compute thin QR factorization $\tilde{\mathbf{B}}_{i,k} = \mathbf{QR}$ and set $\mathbf{B}_{i,k}^+ \leftarrow \mathbf{Q}$.

5: **return** Updated basis $\mathbf{B}_{i,k}^+$ satisfying $\mathbf{B}_{i,k}^{+\top}\mathbf{B}_{i,k}^+ = I$ and $\mathbf{B}_{i,k}^{+\top}\mathbf{B}_{i,q} = 0$ for all $q < k$ (up to numerical error).

---

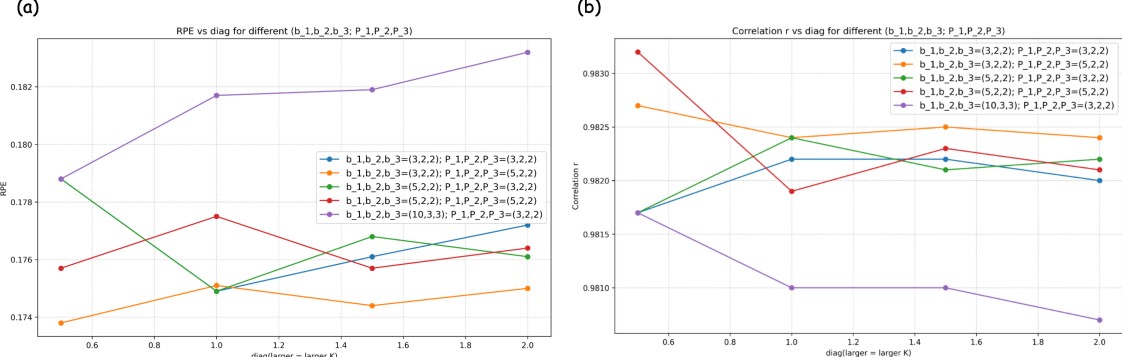

Figure 5: Grid search in rank

| Symbol | Type | Description |
|---|---|---|
| $s$ | integer index | Sample index, $s = 1, \ldots, N$. |
| $N$ | scalar | Number of samples. |
| $n$ | scalar | Number of input modes of $X$. |
| $m$ | scalar | Number of output modes of $Y$. |
| $d_i$ | scalar | Input dimension of mode $i$, $i = 1, \ldots, n$. |
| $o_j$ | scalar | Output dimension of mode $j$, $j = 1, \ldots, m$. |
| $[K] = \{1, \cdots, K\}$ | set | Abbreviation for the set consisting of elements from 1 to K. |
| **Data and regression model** | | |
| $\mathcal{X} \in \mathbb{R}^{N \times d_1 \times \cdots \times d_n}$ | tensor | Input tensor of all $N$ samples. |
| $\mathcal{Y} \in \mathbb{R}^{N \times o_1 \times \cdots \times o_m}$ | tensor | Output tensor of all $N$ samples. |
| $\mathcal{X}_s \in \mathbb{R}^{d_1 \times \cdots \times d_n}$ | tensor | Input tensor of sample $s$. |
| $\mathcal{Y}_s \in \mathbb{R}^{o_1 \times \cdots \times o_m}$ | tensor | Output tensor of sample $s$. |
| $\mathcal{B}$ | tensor | Coefficient tensor of the tensor-on-tensor (TOT) regression model. |
| **IMOBT structure** | | |
| $K$ | scalar | Number of components / blocks. |
| $\mathcal{K} = 1, \cdots, K$ | Index set | Index set of components. |
| $\mathbf{p}$ | multi-index | Component index, e.g. $\mathbf{p} = (p_1, \ldots, p_n)$. |
| $\mathcal{P}$ | index set | Set of active component tuples. |
| $b_i$ | scalar | Subspace dimension of mode $i$. |
| $r_i$ | scalar | Effective projection dimension of mode $i$. |
| $\mathbf{B}_{i,p_i} \in \mathbb{R}^{d_i \times b_i}$ | matrix | Mode-$i$ basis (factor) matrix for component k with combination $\mathbf{p}(\mathbf{k})$, with orthonormal columns. |
| $\mathbf{A}_{i,k} \in \mathbb{R}^{b_i \times r_i}$ | matrix | Synergy matrix for component $k$ with combination $\mathbf{p}(\mathbf{k})$ on mode $i$, with orthonormal columns. |
| $\mathbf{M}_{i,p_i} = \mathbf{B}_{i,p_i} \mathbf{A}_{i,k} \in \mathbb{R}^{d_i \times r_i}$ | matrix | Effective projection matrix on mode $i$ for component $k$ with combination $\mathbf{p}(\mathbf{k})$. |
| $\mathcal{X}_{\text{core}}^{(k)} \in \mathbb{R}^{N \times r_1 \times \cdots \times r_n}$ | tensor | Core tensor obtained by projecting $X$ with $\{M_{i,p_i}^\top\}_{i=1}^n$ for component $p$. |
| $\mathcal{X}_c \in \mathbb{R}^{N \times r_1 \times \cdots \times r_n \times |\mathcal{P}|}$ | tensor | Concatenation of all core tensors along the component mode. |
| $\mathcal{W}_k \in \mathbb{R}^{r_1 \times \cdots \times r_n \times o_1 \times \cdots \times o_m}$ | tensor | Regression head for component $k$. |
| $\mathcal{W} \in \mathbb{R}^{r_1 \times \cdots \times r_n \times |\mathcal{P}| \times o_1 \times \cdots \times o_m}$ | tensor | Regression head on the concatenation of all components. |
| $U_k$ | composite module | denotes a collection of $B_{i,p_i}$-combinations used to extract the principal components. |
| **Manifold** | | |
| $\mathcal{M}_{i,p_i}$ | manifold | Feasible set of $B_{i,p_i}$ (Stiefel with cross-component orthogonality). |
| $\text{St}(d,b)$ | manifold | Stiefel manifold of $d \times b$ matrices with orthonormal columns. |
| **Operator and Linear algebra** | | |
| $\mathcal{N}(0, \Sigma)$ | distribution | Multivariate normal with mean 0 and covariance $\Sigma$. |
| $\Sigma_i$ | matrix | Mode-$i$ covariance matrix of the tensor-variate normal design. |
| $\otimes$ | operator | Kronecker product. |
| $\times_i$ | operator | Mode-$i$ tensor–matrix product. |
| $\langle A, B \rangle$ | scalar | Frobenius inner product between matrices/tensors. |
| $\|A\|_F$ | scalar | Frobenius norm of $A$. |
| $\|a\|_2$ | scalar | Euclidean norm of vector $a$. |
| $\mathbf{I}_d$ | matrix | $d \times d$ identity matrix. |

Table 3: Summary of notation.

