# OpenReview forum: "OSCAR: Orthogonalized Sequential Component Analysis for Tensor-on-Tensor Regression"
_ICLR.cc/2026/Conference — Submitted to ICLR 2026_

### Official Review · Reviewer_Xa7y · 2025-10-30

**Soundness:** 1
**Presentation:** 1
**Contribution:** 2
**Rating:** 2
**Confidence:** 3

**Summary:**

This paper addresses the tensor-on-tensor regression problem by introducing OSCAR, an algorithm that tries to avoid the pitfalls of low-rank approximations and deflation-based sequential methods. The key idea is an Input-Mode Orthogonal Block Term (IMOBT) structure that yields supervised, mutually orthogonal components in the coefficient tensor. Optimization is performed via a Sequential Riemannian Optimization. There is an additional 'collaborative refinement step' that re-optimizes all previously learned components whenever a new one is added. Experiments on synthetic and real datasets are presented.

**Strengths:**

The challenges the paper aims to address are clearly stated in the Introduction. The representation of the tensor regression coefficient is novel.

**Weaknesses:**

Some acronyms in the Introduction are not expanded on first appearance (e.g., N-PLS, HOPLS).

The notation is inconsistent and occasionally confusing. For example, ``Mode-$n$ unfolding'' defined on p. 3 is never used later. The loss is denoted both by $L$ and $\mathcal{L}$.

Several symbols appear without definition or with unclear roles, e.g., $b_i$ (p. 5, first paragraph), $r_i$ (p. 5, second paragraph), and $\Theta$ in Eq. (11). In addition, the ``population risk'' on p. 13 includes a regularizer on $W$ without justification for why $W$ should be penalized at that stage.

The paper discusses ``low-rank'' and ``sparsity'' but does not formally define the low-rankness imposed (e.g., Tucker rank, CP rank, block-term ranks) or motivate the sparsity pattern (which factors/entries are encouraged to be sparse and why).

The algorithm is not fully presented. There is no information about how A and W are optimized in section 3. There is only vague description of the order of optimization of $B_{i,p}$, however, the model has parameters W, A, B jointly generating the regression coefficient.

Key settings are missing. For several experiments, the number of components $K$ is not reported; only Fig.~4 specifies $K=4$. Please report $K$, rank-related hyperparameters, regularization strengths, and optimization details (learning rate, epochs/iterations, tolerances) for all experiments.

In Fig. 4, the first principal component outperforms baselines, yet adding the 2nd--4th components yields worse overall performance than existing methods. This trend raises concerns about the results in the previous experiments. Please (i) analyze why later components hurt performance (e.g., overfitting, non-orthogonality in practice, suboptimal refinement), (ii) report per-component contributions with confidence intervals, and (iii) clarify whether this indicates the method is overall inferior to baselines when multiple components are used.

**Questions:**

See "Weaknesses".

---

> ### Author Response · Authors · 2025-11-23
> **Author Response**
>
> We thank reviewer (comments marked in orange) for the careful and constructive feedback. In the revised manuscript, we have expanded all acronyms at first appearance, cleaned up and unified the notation (with a notation table), clarified the definitions of key symbols and the motivation for the regularizer, refined the description of the optimization procedure, added the missing hyperparameter details, and clarified the interpretation of Figure 4. Below we respond to each comment point by point.
> **Q1:Some acronyms in the Introduction are not expanded on first appearance (e.g., N-PLS, HOPLS).**
>
> **A1:**
> In therevised version, we have checked all acronyms and ensured that each one is expanded at its first appearance in the Introduction.
>
> **Q2:The notation is inconsistent and occasionally confusing. For example, "Mode-unfolding" defined on p. 3 is never used later. The loss is denoted both by $L$ and $\mathcal{L}$.**
>
> **A2:**
> We apologize for the resulting confusion. In the revised version, we have unified the notation for the loss function and consistently use $\mathcal{L}$ throughout the paper. We have also fixed the issue with the definition of mode-unfolding on p.3:the notation is now either aligned with its later usage or removed if not needed. In addition, we have added a notation table to the paper to ensure global consistency and to make the symbols easier to follow for the reader.
>
> **Q3: Several symbols appear without definition or with unclear roles, e.g. $b_i$, $r_i$, and $\Theta$ in Eq. (11). In addition, the "population risk" includes a regularizer on $W$ without justification for why $W$ should be penalized.**
>
> **A3:**
> We clarify the notational issues and the role of the regularizer.
>
> *1、Definitions of $b_i$, $r_i$, and $\Theta$.*
> In the revised version, we explicitly restate that the definitions and roles of $b_i$, $r_i$, and $\Theta$ are given in Section3.2 (Input-Mode Orthogonal Block Term (IMOBT) Decomposition). These symbols are now also included and clearly explained in the notation table to ensure consistency and ease of reference throughout the paper.
>
> *2、Regularizer on $W$ in the population risk.*
> The inclusion of a regularization term on $W$ in the population risk follows standard statistical practice. Specifically, the $\ell_2$ penalty is equivalent to ridge regularization, which helps alleviate overfitting caused by the fully connected layer and stabilizes the optimization landscape. This justification has been added to the revised manuscript to make the motivation clear.
>
> **Q4:The algorithm is not fully presented. There is no information about how A and W are optimized in section 3. There is only vague description of the order of optimization of, however, the model has parameters W, A, B jointly generating the regression coefficient.**
>
> **A4:**
> We thank the reviewer for pointing out that the optimization procedure was not sufficiently clear in the original submission. In the revised manuscript, we have now made the optimization process explicit in the main text. We have $\textbf{added Algorithm 1 and Algorithm 2}$, which provide step-by-step pseudo-code for updating $\mathbf{A}$,$\mathbf{B}$ and $\mathbf{W}$, and we have revised the flowchart in Fig.2 to visualize the overall optimization pipeline more clearly. In addition, we have substantially revised the discussion in Section 3 to describe the order of the updates and the joint optimization in a more self-contained and easy-to-follow manner.
>
> **Q5:Key settings are missing.**
>
> **A5:**
> We apologize for not fully reporting the hyperparameter settings and optimization configurations in the initial submission. In the revised manuscript, we have added all rank-related hyperparameters, regularization strengths, and optimization details in Appendix H.
>
> **Q6:In Fig. 4, the first principal component outperforms baselines, yet adding the 2nd--4th components yields worse overall performance than existing methods. This trend raises concerns about the results in the previous experiments.**
>
> **A6:**
> We sincerely apologize for the confusion caused by the caption and description of Fig. 4. We clarify that $\textbf{Figure 4 depicts the predictive performance of each individual component independently}$ (evaluated via a separate linear probe), rather than the $\textbf{cumulative}$ performance of the model as components are added. We have revised the manuscript to make this distinction clear, and the submitted code also confirms that the components are evaluated independently.
>
> Under this interpretation, the results highlight a key strength of our method: OSCAR’s first principal component achieves a substantially lower RPE than HOPLS, validating our $\textbf{Cooperative Refinement Mechanism}$ and $\textbf{Orthogonality Constraints}$. We further repeated the per-component evaluation over multiple random seeds and added confidence intervals; the consistent patterns across trials demonstrate that the effect is robust and not due to randomness.

---

> ### Author Response · Authors · 2025-11-27
> **Looking forward to your feedback**
>
> Dear Reviewer Xa7y,
>
> We are eager to ensure that we have adequately addressed your concerns and are prepared to offer further clarifications or address any additional questions you may have.
>
> Should you find that our revisions have satisfactorily addressed your main concerns, we would be most grateful if you would **reconsider the evaluation of our paper to enhance its standing**.
>
> We would like to express our heartfelt gratitude for the time and effort you have dedicated to reviewing our work. It has been a pleasure to engage with you throughout this process.😁
>
> Best regards,
>
> The authors

---

### Official Review · Reviewer_Z8Sz · 2025-11-01

**Soundness:** 3
**Presentation:** 1
**Contribution:** 2
**Rating:** 4
**Confidence:** 3

**Summary:**

In this paper, the authors develop an OSCAR framework for tensor-on-tensor (TOT) regression. The IMOBT structure is introduced for supervised extraction of orthogonal components. The Riemannian optimization is utilized to enforce orthogonality in parameter space, and a refinement mechanism is utilized to get the global solutions. Theoretical analysis of the proposed algorithm is provided. Experiments on synthetic and real-world datasets show the desired performance of the proposed method.

**Strengths:**

1. A new framework for ToT is developed.
2. The IMOBT and SRO are proposed to enable orthogonal component extraction and enforce orthogonality.
3. Experimental results show the improvement of the proposed method compared to existing algorithms.

**Weaknesses:**

1. The paper is not well organized. The main algorithm and its theoretical analysis are not shown in the main paper.
2. The time complexity of the proposed method is not given.
3. Experiments are limited to 3-order tensors.

**Questions:**

1. The organization of the paper should be improved. The pseudo-code should be provided. The main algorithm should appear in the main text, not in the appendix. More explanation for Figure 2 should be added.
2. The appendices are hard to read. For example, it is unclear what Appendix A and Theorem 1 derive for (I cannot find any relation in the paper to Appendix A or Theorem 1). Also, it is unclear which algorithm the convergence guarantee in Appendix B refers to (perhaps the algorithm in Appendix C).
3. What is the theoretical advantage compared to TTReg (Qin & Zhu (2025))?
4. What is the time complexity of the proposed method, and how does its running time compare to other methods?
5. How does the method perform on higher-order tensors (order 4 or higher)?
6. Some typos are not well described. What does the y label (R^2) in Figure 3 mean? What does “population model” mean in the paper (it appears in line 339/547)?

---

> ### Author Response · Authors · 2025-11-23
> **Author Response**
>
> We thank Reviewer 2 (red) for the insightful comments. We have carefully revised the manuscript, including reorganizing the main algorithm and theory into the main text, providing explicit time-complexity analysis and pseudo-code, extending the experiments, and clarifying the theoretical contributions and terminology. Below we respond to each point in detail.
>
> **Q1:The paper is not well organized. The main algorithm and its theoretical analysis are not shown in the main paper.**
>
> **A1:**
> We thank the reviewer for pointing out these organizational issues and apologize for the confusion. In the revision, we have reorganized the paper so that the main algorithmic and theoretical contributions are clearly presented in the main text section 3.
>
> We hope this reorganization makes the structure of the paper clearer and the connection between the algorithm and its theoretical analysis much easier to follow.
>
> **Q2:The time complexity of the proposed method is not given.**
>
> **A2:**
> For dense input tensors of size $D=\prod_{i=1}^n d_i$ and $N$ samples.
> Excluding unavoidable data–access cost, one RGD step in each method can be written as$$T_{\text{step}}= \mathcal{O}\bigl(ND \cdot c(\text{rank parameters})\bigr),$$with rank–dependent constants:$$c_{\mathrm{OSCAR-IMOBT}}= K\sum_{i=1}^n \frac{r_i}{d_i}$$, which is the same order as that of one iteration of TuckerReg[1].
>
> **Q3:Experiments are limited to 3-order tensors.**
>
> **A3:**
> We have supplemented the synthetic-data experiments with an ablation study on tensors of different orders. As shown in Figure 3 of the main text, the proposed method achieves high R-squared values across these settings, demonstrating that it can be effectively fitted to tensors beyond the third order.
>
> **Q4:The pseudo-code should be provided. The main algorithm should appear in the main text. More explanation for Figure 2 should be added.**
>
> **A4:**
> We thank the reviewer for these helpful suggestions and apologize for the confusion caused by the previous organization. In the revised version, we have reorganized the paper to improve the overall flow, incorporated the main OSCAR algorithm directly into the main text, and provided explicit pseudo-code for the full procedure. We have also added more detailed explanation and discussion around Figure 2 to clarify its role and how it supports our claims.
>
> **Q5: The appendices are hard to read. For example, it is unclear what Appendix A and Theorem 1 derive for . It is unclear which algorithm the convergence guarantee in Appendix B refers to.**
>
> **A5:**
> We have included a revised and more detailed version of the derivations in the appendix. This part mainly establishes the theoretical guarantees for our model, including its identifiability as well as the characterization of the region in which the proposed estimator is optimal.
>
> **Q6:What is the theoretical advantage compared to TTReg ?**
>
> **A6:**
> Theoretical guarantees in TTReg.TTReg studies tensor-on-tensor regression under a low TT-rank assumption and shows that, under an RIP-type condition, sufficiently many samples, and a good spectral initialization close to the truth, their IHT/RGD algorithms enjoy "local linear convergence" to a small neighborhood of the true parameter. The paper does not provide global convergence from arbitrary initializations.
>
> Our theoretical guarantee.In Appendix A, we prove that under our model assumptions the estimated regression tensor converges to the true parameter. In addition, our framework naturally yields multiple principal components that are “orthogonal across multiple dimensions” by construction, which leads to strong feature disentanglement and constitutes a key advantage over TTReg and related tensor regression methods.
>
>
> **Q7:What does the y label ($R^2$) in Figure 3 mean? What does “population model” mean in the paper ?**
>
> **A7:**
> *Meaning of the $y$-axis label "$R^2$" in Figure 3.*
> We apologize for the lack of clarity. The $y$-axis label $R^2$" denotes the coefficient of determination, which measures the proportion of variance in the response that is explained by the model. We have already provided its definition and mathematical expression in Section 4.1.1, and we will make this connection clearer in the revised version.
>
> *Meaning of "population model".*
> In statistics, a "population model" refers to a parametric mathematical equation describing the data-generating mechanism at the population level, i.e., it specifies the deterministic relationships between variables together with the structure of the unobservable random errors. In our paper, the term "population model" specifically refers to the tensor-on-tensor regression model defined under our proposed IMOBT-based structural assumptions. We will clarify this terminology in the revised manuscript to avoid confusion.
>
> **Reference**
>
> [1]. Llosa-Vite, C., & Maitra, R. (2022). Reduced-rank tensor-on-tensor regression and tensor-variate analysis of variance.

---

> ### Author Response · Authors · 2025-11-27
> **Looking forward to your feedback**
>
> Dear Reviewer Z8Sz,
>
> We are eager to ensure that we have adequately addressed your concerns and are prepared to offer further clarifications or address any additional questions you may have.
>
> Should you find that our revisions have satisfactorily addressed your main concerns, we would be most grateful if you would **reconsider the evaluation of our paper to enhance its standing**.
>
> We would like to express our heartfelt gratitude for the time and effort you have dedicated to reviewing our work. It has been a pleasure to engage with you throughout this process.😁
>
> Best regards,
>
> The authors

---

### Official Review · Reviewer_5EqZ · 2025-11-03

**Soundness:** 3
**Presentation:** 1
**Contribution:** 2
**Rating:** 4
**Confidence:** 3

**Summary:**

To remedy the issue of  “curse of dimensionality” arising in tensor-on-tensor regression, this work proposes a novel framework that unifies low-rank modeling with supervised orthogonal component extraction. This framework could also enable the supervised extraction of orthogonal components, resulting in a more accurate and interpretable regression framework. Extensive experiments on both synthetic and real-world datasets are carried out to show that the proposed framework not only surpasses existing state-of-the-art methods in predictive performance but also has unique advantages in model interpretability and feature disentanglement.

**Strengths:**

1. A soundness and efficient framework is provided to address the issue of  “curse of dimensionality” for the tensor-on-tensor problem.
2. A new stage-wise RGD-based optimization scheme is developed to solving the resulting problem.
3. Extensive experiments are carried out to demonstrate the merits of the proposed framework.

**Weaknesses:**

1. The procedure for generating synthetic data is not clearly described.
2. The proof sketch of the main Theorem 1 is hard to follow.
3. The reasonableness of the Assumptions 1-5 was not discussed.

**Questions:**

1. How to choose the starting point for the developed RGD algorithm?
2. How does the performance of the proposed framework extend beyond the setting of Gaussian noise?
3. How can the tensor rank be effectively tuned for the proposed framework in practice?

---

> ### Author Response · Authors · 2025-11-23
> **Author Response**
>
> We thank the reviewer for the insightful comments. The experiments on synthetic data and the theoretical proofs of the model have been included in the resubmitted manuscript. We have highlighted in blue the main revisions made in response to this reviewer’s comments and will carefully address the remaining issues in the revised version.
>
> **Q1:The procedure for generating synthetic data is not clearly described.**
>
> **A1:**
> We have provided a clear description in Appendix E.1.
> The procedure for generating synthetic data contains the following steps:
> 1. We generate a random coefficient tensor $\mathcal{B}^\star$.
> 2. We approximate $\mathcal{B}^\star$ by IMOBT structure using an alternating least squares (ALS) procedure to generate true low-rank coefficient tensor $\widetilde{\mathcal{B}}$.
> 3. $X$ are generated with independently Gaussian distribution.
> 4. $Y$ are generated by applying $\widetilde{\mathcal{B}}$ to $X$ and adding Gaussian noise.
>
> **Q2:The proof sketch of the main Theorem 1 is hard to follow.**
>
> **A2:**
> We have thoroughly revised Theorem 1 and the proof to ensure all notation is clear, standardized, and used consistently throughout the paper. The proof sketch is revised in Appendix B.
>
>
> **Q3:The reasonableness of the Assumptions 1-5 was not discussed.**
>
> **A3:**
> We agree that the reasonableness of Assumptions 1–5 is important and have added a detailed discussion in Appendix B.6. Here we give a brief overview. Assumption 1 adopts the standard Gaussianity and independence conditions for the covariates and noise that are widely used in tensor regression [3]. Assumptions 2–3 posit that the true model has an IMOBT-type low-rank structure with orthonormal mode-wise projection matrices, in line with common structural low-rank assumptions in tensor regression. Assumption 4 is a standard signal-strength (beta-min) condition requiring active blocks to have non-negligible regression weights so they are distinguishable from noise [1]. Finally, Assumption 5 is an identifiability condition ensuring that, up to permutations and orthogonal rotations within each latent subspace, the population IMOBT representation is unique, analogous to Kruskal-type identifiability results for block-term decompositions [2].
>
> **Q4:How to choose the starting point for the developed RGD algorithm?**
>
> **A4:**
> We initialize the proposed RGD algorithm using an orthogonal basis constructed from the discrete Fourier transform (DFT), rather than with a random initialization. This choice allows the starting point to emphasize low-frequency components, which generally contain the most significant structural information in the data, and therefore provides a more informative and stable initialization for subsequent optimization.We added this in the revised manuscript in Appendix D.3.
>
> **Q5:How does the performance of the proposed framework extend beyond the setting of Gaussian noise?**
>
> **A5:**
> Our theoretical analysis and simulations adopt the classical assumption that the regression errors are Gaussian and independent of the covariates. This least-squares–type setting underlies classical linear regression theory and most existing tensor and tensor-on-tensor (TOT) regression work [3][4], and our aim is to study optimization and statistical properties within this standard Gaussian TOT framework.
>
> In particular, we add Gaussian noise to the responses following the experimental setup in [5], so that our robustness evaluation is directly comparable to recent methods. Extending the framework and theory to genuinely non-Gaussian or heavy-tailed noise is an interesting direction for future work, but lies outside the main scope of the present paper.
>
> **Q6:How can the tensor rank be effectively tuned for the proposed framework in practice?**
>
> **A6:**
> We appreciate the reviewer’s question on practical implementation. In our experiments, the component sizes and tensor ranks are mainly guided by the physical dimensions of the input matrices X and Y and by the desired number of principal components. For general applications, we recommend selecting the ranks via a simple grid search within the bounds imposed by the data dimensions. As shown in Fig. 5 of the appendix, exploring a moderate range of rank configurations consistently reveals an *appropriate operating region* where model capacity and reconstruction error are well-balanced without introducing substantial computational overhead.
>
> **Reference**
>
> [1]. Bühlmann, P., & Van De Geer, S. (2011). Statistics for high-dimensional data: methods, theory and applications.
>
> [2]. De Lathauwer, L. (2008). Decompositions of a higher-order tensor in block terms—Part II: Definitions and uniqueness.
>
> [3]. Llosa-Vite, C., & Maitra, R. (2022). Reduced-rank tensor-on-tensor regression and tensor-variate analysis of variance.
>
> [4]. Lock, E. F. (2018). Tensor-on-tensor regression.
>
> [5]. Qin, Z., & Zhu, Z. (2025). Computational and statistical guarantees for tensor-on-tensor regression with tensor train decomposition.

---

> ### Author Response · Authors · 2025-11-27
> **Looking forward to your feedback**
>
> Dear Reviewer 5EqZ,
>
> We are eager to ensure that we have adequately addressed your concerns and are prepared to offer further clarifications or address any additional questions you may have.
>
> Should you find that our revisions have satisfactorily addressed your main concerns, we would be most grateful if you would **reconsider the evaluation of our paper to enhance its standing**.
>
> We would like to express our heartfelt gratitude for the time and effort you have dedicated to reviewing our work. It has been a pleasure to engage with you throughout this process.😁
>
> Best regards,
>
> The authors

---

### Author Response · Authors · 2025-11-25
**General Response to All Reviewers**

Dear Reviewers,

We sincerely thank all the reviewers (5EqZ, Z8Sz, Xa7y) for their valuable and constructive feedback. We are encouraged that you found the proposed framework sound and meaningful (5EqZ, Z8Sz), and that you appreciated the empirical evaluation and comparisons (5EqZ, Z8Sz, Xa7y), while also providing helpful suggestions on improving the organization and clarity of the presentation (5EqZ, Z8Sz, Xa7y).

We have carefully addressed all comments in the point-by-point responses. In particular, we have:
- Clarified the problem setup, notation, and assumptions, and reorganized the main sections to present the model and key theorem more clearly (5EqZ, Z8Sz, Xa7y).
- Added implementation details and pseudo-code for the proposed algorithm, including the initialization strategy, time-complexity analysis, and practical guidelines for choosing ranks and other hyperparameters (5EqZ, Z8Sz).
- Expanded and clarified the experiments with additional studies and ablations (e.g., on higher-order tensors), and improved the explanations of metrics, figures, and tables, as well as reporting full hyperparameter and optimization settings (5EqZ, Z8Sz, Xa7y).
- Strengthened the discussion of related work and the reasonableness of our assumptions, and clarified the positioning of our method relative to existing tensor-on-tensor regression approaches (5EqZ, Z8Sz).

In the $\textbf{revised manuscript}$:
- Changes that address $\textbf{comments common to all reviewers}$ are highlighted in $\textbf{green}$.
- Changes mainly addressing $\textbf{Reviewer 5EqZ}$ are highlighted in $\textbf{blue}$.
- Changes mainly addressing $\textbf{Reviewer Z8Sz}$ are highlighted in $\textbf{red}$.
- Changes mainly addressing $\textbf{Reviewer Xa7y}$ are highlighted in $\textbf{orange}$.

We hope these revisions adequately address your concerns and further improve the quality and readability of the paper. If you find the revisions satisfactory, we would be very grateful if you could consider an improved score.

Best regards,

Authors

---

### Author Response · Authors · 2025-12-02
**Summary of the Discussion(1/3)**

Dear ICLR 2026 AC, SAC, PC and Reviewers,

We would like to sincerely thank you and all the reviewers (`5EqZ`, `Z8Sz`, `Xa7y`) for the time and effort devoted to evaluating our submission **“OSCAR: Orthogonalized Sequential Component Analysis for Tensor-on-Tensor Regression”**. We greatly appreciate your constructive feedback on both the technical aspects and the presentation of the work.

We have carefully revised the manuscript and rebuttal in response to all comments. We would be extremely grateful if you could kindly check whether the current version of the paper adequately addresses the main concerns raised in the reviews. To facilitate your assessment, we summarize below the key strengths highlighted by the reviewers and the corresponding revisions we have made to address the identified weaknesses.

## Strengths

### 1. Sound & efficient framework for the “curse of dimensionality” (Reviewer `5EqZ`)
Reviewer `5EqZ` highlighted that OSCAR provides a sound and efficient framework to address the “curse of dimensionality” in the tensor-on-tensor problem. Motivated by this feedback, we aim for the framework to be broadly applicable to high-dimensional tensor-on-tensor regression settings.

### 2. New framework for ToT regression (Reviewer `Z8Sz`)
Reviewer `Z8Sz` noted that a new framework for ToT regression is developed. We interpret this as supporting our intent to propose a general modeling and learning pipeline that can be adapted to different tensor regression scenarios.

### 3. Orthogonal component extraction via IMOBT & SRO (Reviewer `Z8Sz`)
Reviewer `Z8Sz` stated that IMOBT and SRO enable orthogonal component extraction and enforce orthogonality. Based on this comment, we hope the orthogonality-enforced sequential extraction can be useful for learning disentangled and interpretable components.

### 4. Stage-wise RGD-based optimization scheme (Reviewer `5EqZ`)
Reviewer `5EqZ` highlighted the development of a new stage-wise RGD-based optimization scheme for solving the resulting problem. Following this feedback, we further improved the presentation of the optimization procedure to make it easier to reproduce and extend.

### 5. Extensive experiments & improvements over baselines (Reviewers `5EqZ`, `Z8Sz`)
Reviewer `5EqZ` emphasized that extensive experiments demonstrate the merits of the proposed framework, and reviewer `Z8Sz` noted that experimental results show improvement compared to existing algorithms. Encouraged by these comments, we strengthened the experimental reporting and clarified evaluation details where requested.

### 6. Clear motivation & novel coefficient representation (Reviewer `Xa7y`)
Reviewer `Xa7y` stated that the challenges are clearly presented in the Introduction, and that the representation of the tensor regression coefficient is novel. We are grateful for this recognition and further refined the exposition around the motivation and coefficient parameterization.

---

> ### Author Response · Authors · 2025-12-02
> **Summary of the Discussion(2/3)**
>
> ## Weaknesses and Revisions
>
> We summarize the main concerns raised by reviewers, along with the corresponding revisions in the current manuscript.
>
> ### 1. Interpretation of experiments and Figure 4 (`Xa7y`)
>
> **Concern:** In Figure 4, the first principal component outperforms baselines, yet adding the 2nd–4th components appeared to yield worse overall performance than existing methods; this raised concerns about consistency with earlier experiments. Reviewer `Xa7y` asked us to (i) analyze why later components hurt performance, (ii) report per-component contributions with confidence intervals, and (iii) clarify whether the method is overall inferior when multiple components are used.
>
> **Revisions:**
> We clarified that Figure 4 evaluates each component individually via a linear probe—each component is evaluated separately using an $\ell_2$-regularized FC layer—rather than reporting cumulative performance when stacking multiple components. We added the requested analysis by reporting per-component behavior with repeated runs and confidence intervals, expanded the discussion of why Figure 4 does not imply that adding components degrades overall OSCAR performance, and revised the caption and surrounding text to prevent misinterpretation.
>
> ### 2. Clarity of synthetic data generation and experimental setup (`5EqZ`, `Z8Sz`)
>
> **Concern:** The synthetic data generation process was not clearly described, and experiments seemed limited to 3rd-order tensors.
>
> **Revisions:**
> We added a detailed step-by-step description of the synthetic data generation pipeline in **Appendix E.1**, including how the ground-truth coefficient tensor is constructed, approximated by IMOBT via ALS, and how $X$ and $Y$ are subsequently sampled. We also expanded the synthetic experiments to include higher-order tensors and provided ablations on the number of components, feature extraction dimensions, and tensor order.
>
> ### 3. Accessibility of the theory and discussion of assumptions (`5EqZ`, `Z8Sz`)
>
> **Concern:** The proof sketch of Theorem 1 was hard to follow, and the reasonableness of Assumptions 1–5 and terms like “population model” were not clearly discussed.
>
> **Revisions:**
> We revised Theorem 1 and its proof in **Appendix B**, standardizing notation and improving the structure so the proof flow and connections to the main text are clearer. We added **Appendix B.6** to discuss the reasonableness of Assumptions 1–5 and clarified the meaning of the “population model”.
>
> ### 4. Missing algorithmic details, optimization order, and time complexity (`5EqZ`, `Z8Sz`, `Xa7y`)
>
> **Concern:** The optimization procedure for $A$, $B$, and $W$ was not fully specified, the order of updates was only vaguely described, and time complexity and practical guidance on rank selection and initialization were missing.
>
> **Revisions:**
> We added explicit pseudo-code and clarified the update order for $B$, $A$, and $W$, including how sequential refinement is performed. We documented the initialization strategy and provided a time-complexity discussion (e.g., for dense inputs of size $D=\prod_i d_i$ and $N$ samples, one step cost can be written as $T_{\text{step}}=\mathcal{O}(ND\cdot c(\text{rank parameters}))$ with rank-dependent constants). We also added practical guidelines for choosing ranks and centralized hyperparameter/optimization settings in the appendix.
>
> ### 5. Notation, acronyms, and presentation consistency (`Z8Sz`, `Xa7y`)
>
> **Concern:** Several acronyms were not expanded on first appearance; notation was inconsistent (e.g., loss denoted by both $L$ and $\mathcal{L}$); and multiple symbols (e.g., $b_i$, $r_i$, $\Theta$) lacked clear definitions.
>
> **Revisions:**
> We expanded acronyms at first mention and unified symbols and notation. We added a notation table, clarified the roles/definitions of $b_i$, $r_i$, $\Theta$, and standardized the loss notation (using $\mathcal{L}$ consistently). We also justified why a regularizer on $W$ appears in the population risk discussion and refined the writing for clarity.
>
> ### 6. Theoretical positioning relative to TTReg and other baselines (`Z8Sz`)
>
> **Concern:** The relationship and theoretical positioning relative to TTReg and other tensor regression baselines were not sufficiently clear.
>
> **Revisions:**
> We strengthened the related-work discussion and clarified how OSCAR differs from TTReg and other low-rank ToT methods, both in modeling assumptions and in the sequential orthogonal component extraction perspective.

---

> > ### Author Response · Authors · 2025-12-02
> > **Summary of the Discussion(3/3)**
> >
> > ## Misunderstandings and Clarifications
> >
> > We highlight a few points where our revised manuscript aims to prevent further confusion:
> >
> > 1. **Interpretation of Figure 4 and component evaluation (`Xa7y`)**
> > Figure 4 reports per-component predictive power using a linear probe (each component evaluated separately with an $\ell_2$-regularized FC layer), rather than the cumulative performance of combining multiple components. We added the requested analysis (including repeated runs/confidence intervals) and updated the caption/text to ensure this figure is not misread as evidence that adding components necessarily degrades overall performance.
> >
> > 2. **Algorithm completeness and optimization order**
> > We clarified how parameters $A$, $B$, and $W$ are optimized, the order of updates, and how the sequential refinement mechanism is implemented in practice, with pseudo-code and consolidated training details.
> >
> > 3. **Notation and assumptions**
> > We unified notation (e.g., $\mathcal{L}$) and added a notation table. We also expanded the discussion of assumptions and clarified terminology such as the “population model”.
> >
> >
> >
> > We hope that these clarifications, together with the substantial revisions to the manuscript, address the concerns raised by the reviewers and improve both the technical completeness and the readability of our work. We would be deeply grateful if, after reviewing the revised manuscript and rebuttal, you find it appropriate to reconsider the evaluation of our submission.
> >
> > Thank you again to the Program Chairs, Senior Area Chairs, Area Chairs, and Reviewers for your time, thoughtful feedback, and service to the ICLR community.
> >
> > Best regards,
> >
> > **The authors of submission 3429 (OSCAR)**

---

### Meta-Review · Area_Chair_GqJA · 2025-12-20

**Summary:**

To remedy the issue of “curse of dimensionality” arising in tensor-on-tensor regression, this work proposes a novel framework that unifies low-rank modeling with supervised orthogonal component extraction. However, this paper is not well written and there exist many writing issues which caused many confusions. As pointed out by the reviewers, the paper needs significant reorganization and the clarity needs to be significantly improved, for example, the main algorithm is not shown in the main paper, the proof of theorem 1 is hard to follow, the presentation of the appendix is hard to read, etc. Overall, this requires a significant amount of work to do a major revision, and it is impossible for the reviewers to check the details of the revision in a short time. The authors are urged to improve the presentation when this work is submitted to a future venue.

**Reviewer Concerns:**

Addressed:

1. How to choose starting point of RGD.
2. Procedure of generating synthetic data.

Still outstanding:

1. Clarity.
2. How to choose rank efficiently. Grid search is not a practical idea for large-scale tensors.

**Reviewer Scores:**

The reviewers might agree with some of the reorganization and improved clarity. But since there are too many issues with clarity and reorganization, I feel that it is almost impossible for the reviewers to check all the revised details, and it is thus hard for the reviewers to change their mind to fully agree with the presented contents. The grid search for choosing rank in practice doesn't sound feasible.

---

### Decision · Program_Chairs · 2026-01-26

Reject